# A Tale of Two Flows: Cooperative Learning of Langevin Flow and Normalizing Flow Toward Energy-Based Model

**Jianwen Xie, Yaxuan Zhu, Jun Li, Ping Li**

Cognitive Computing Lab, Baidu Research
10900 NE 8th St. Bellevue, WA 98004, USA
`{jianwen.kenny, yaxuanzhu.alvin, junli.sh04, pingli98}@gmail.com`

## Abstract

This paper studies the cooperative learning of two generative flow models, in which the two models are iteratively updated based on the jointly synthesized examples. The first flow model is a normalizing flow that transforms an initial simple density to a target density by applying a sequence of invertible transformations. The second flow model is a Langevin flow that runs finite steps of gradient-based MCMC toward an energy-based model. We start from proposing a generative framework that trains an energy-based model with a normalizing flow as an amortized sampler to initialize the MCMC chains of the energy-based model. In each learning iteration, we generate synthesized examples by using a normalizing flow initialization followed by a short-run Langevin flow revision toward the current energy-based model. Then we treat the synthesized examples as fair samples from the energy-based model and update the model parameters with the maximum likelihood learning gradient, while the normalizing flow directly learns from the synthesized examples by maximizing the tractable likelihood. Under the short-run non-mixing MCMC scenario, the estimation of the energy-based model is shown to follow the perturbation of maximum likelihood, and the short-run Langevin flow and the normalizing flow form a two-flow generator that we call CoopFlow. We provide an understating of the CoopFlow algorithm by information geometry and show that it is a valid generator as it converges to a moment matching estimator. We demonstrate that the trained CoopFlow is capable of synthesizing realistic images, reconstructing images, and interpolating between images.

## 1 Introduction and Motivation

Normalizing flows (Dinh et al., 2015; 2017; Kingma & Dhariwal, 2018) are a family of generative models that construct a complex distribution by transforming a simple probability density, such as Gaussian distribution, through a sequence of invertible and differentiable mappings. Due to the tractability of the exact log-likelihood and the efficiency of the inference and synthesis, normalizing flows have gained popularity in density estimation (Kingma & Dhariwal, 2018; Ho et al., 2019; Yang et al., 2019; Prenger et al., 2019; Kumar et al., 2020) and variational inference (Rezende & Mohamed, 2015; Kingma et al., 2016). However, for the sake of ensuring the favorable property of closed-form density evaluation, the normalizing flows typically require special designs of the sequence of transformations, which, in general, constrain the expressive power of the models.

Energy-based models (EBMs) (Zhu et al., 1998; LeCun et al., 2006; Hinton, 2012) define an unnormalized probability density function of data, which is the exponential of the negative energy function. The energy function is directly defined on data domain, and assigns each input configuration with a scalar energy value, with lower energy values indicating more likely configurations. Recently, with the energy function parameterized by a modern deep network such as a ConvNet, the ConvNet-EBMs (Xie et al., 2016) have gained unprecedented success in modeling large-scale data sets (Gao et al., 2018; Nijkamp et al., 2019; Du & Mordatch, 2019; Grathwohl et al., 2020; Gao et al., 2021; Zhao et al., 2021; Zheng et al., 2021) and exhibited stunning performance in synthesizing different modalities of data, e.g., videos (Xie et al., 2017; 2021c), volumetric shapes (Xie et al.,

2018b; 2020b), point clouds (Xie et al., 2021a) and molecules (Du et al., 2020). The parameters of the energy function can be trained by maximum likelihood estimation (MLE). However, due to the intractable integral in computing the normalizing constant, the evaluation of the gradient of the log-likelihood typically requires Markov chain Monte Carlo (MCMC) sampling (Barbu & Zhu, 2020), e.g., Langevin dynamics (Neal, 2011), to generate samples from the current model. However, the Langevin sampling on a highly multi-modal energy function, due to the use of deep network parameterization, is generally not mixing. When sampling from a density with a multi-modal landscape, the Langevin dynamics, which follows the gradient information, is apt to get trapped by local modes of the density and is unlikely to jump out and explore other isolated modes. Relying on non-mixing MCMC samples, the estimated gradient of the likelihood is biased, and the resulting learned EBM may become an invalid model, which is unable to approximate the data distribution as expected.

Recently, Nijkamp et al. (2019) propose to train an EBM with a short-run non-convergent Langevin dynamics, and show that even though the energy function is invalid, the short-run MCMC can be treated as a valid flow-like model that generates realistic examples. This not only provides an explanation of why an EBM with a non-convergent MCMC is still capable of synthesizing realistic examples, but also suggests a more practical computationally-affordable way to learn useful generative models under the existing energy-based frameworks. Although EBMs have been widely applied to different domains, learning short-run MCMC in the context of EBM is still underexplored.

In this paper, we accept the fact that MCMC sampling is not mixing in practice, and decide to give up the goal of training a valid EBM of data. Instead, we treat the short-run non-convergent Langevin dynamics, which shares parameters with the energy function, as a flow-like transformation that we call the Langevin flow because it can be considered a noise-injected residual network. Even though implementing a short-run Langevin flow is surprisingly simple, which is nothing but a design of a bottom-up ConvNet for the energy function, it might still require a sufficiently large number of Langevin steps (each Langevin step consists of one step of gradient descent and one step of diffusion) to construct an effective Langevin flow, so that it can be expressive enough to represent the data distribution. Motivated by reducing the number of Langevin steps in the Langevin flow for computational efficiency, we propose the CoopFlow model that trains a Langevin flow jointly with a normalizing flow in a cooperative learning scheme (Xie et al., 2020a), in which the normalizing flow learns to serve as a rapid sampler to initialize the Langevin flow so that the Langevin flow can be shorter, while the Langevin flow teaches the normalizing flow by short-run MCMC transition toward the EBM so that the normalizing flow can accumulate the temporal difference in the transition to provide better initial samples. Compared to the original cooperative learning framework (Xie et al., 2020a) that incorporates the MLE algorithm of an EBM and the MLE algorithm of a generator, the CoopFlow benefits from using a normalizing flow instead of a generic generator because the MLE of a normalizing flow generator is much more tractable than the MLE of any other generic generator. The latter might resort to either MCMC-based inference to evaluate the posterior distribution or another encoder network for variational inference. Besides, in the CoopFlow, the Langevin flow can overcome the expressivity limitation of the normalizing flow caused by invertibility constraint, which also motivates us to study the CoopFlow model. We further understand the dynamics of cooperative learning with short-run non-mixing MCMC by information geometry. We provide a justification that the CoopFlow trained in the context of EBM with non-mixing MCMC is a valid generator because it converges to a moment matching estimator. Experiments, including image generation, image reconstruction, and latent space interpolation, are conducted to support our justification.

## 2 CONTRIBUTIONS AND RELATED WORK

Our paper studies amortized sampling for training a short-run non-mixing Langevin sampler toward an EBM. We propose a novel framework, the CoopFlow, which cooperatively trains a short-run Langevin flow as a valid generator and a normalizing flow as an amortized sampler for image representation. We provide both theoretical and empirical justifications for the proposed CoopFlow algorithm, which has been implemented in the PaddlePaddle deep learning platform. The following are some closely related work. We will point out the differences between our work and the prior arts to further highlight the contributions and novelty of our paper.

**Learning short-run MCMC as a generator.** Recently, Nijkamp et al. (2019) propose to learn an EBM with short-run non-convergent MCMC that samples from the model, and they eventually keep the short-run MCMC as a valid generator and discard the biased EBM. Nijkamp et al. (2020b)

use short-run MCMC to sample the latent space of a top-down generative model in a variational learning framework. An et al. (2021) propose to correct the bias of the short-run MCMC inference by optimal transport in training latent variable models. Pang et al. (2020) adopt short-run MCMC to sample from both the EBM prior and the posterior of the latent variables. Our work studies learning a normalizing flow to amortize the sampling cost of a short-run non-mixing MCMC sampler (i.e., Langevin flow) in data space, which makes a further step forward in this underexplored theme.

**Cooperative learning with MCMC teaching.** Our learning algorithm is inspired by the cooperative learning or the CoopNets (Xie et al., 2018a; 2020a), in which an ConvNet-EBM (Xie et al., 2016) and a top-down generator (Han et al., 2017) are jointly trained by jump-starting their maximum learning algorithms. Xie et al. (2021b) replace the generator in the original CoopNets by a variational autoencoder (VAE) (Kingma & Welling, 2014) for efficient inference. The proposed CoopFlow algorithm is different from the above prior work in the following two aspects. (i) In the idealized long-run mixing MCMC scenario, our algorithm is a cooperative learning framework that trains an unbiased EBM and a normalizing flow via MCMC teaching, where updating a normalizing flow with a tractable density is more efficient and less biased than updating a generic generator via variational inference as in Xie et al. (2021b) or MCMC-based inference as in Xie et al. (2020a). (ii) Our paper has a novel emphasis on studying cooperative learning with short-run non-mixing MCMC, which is more practical and common in reality. Our algorithm actually trains a short-run Langevin flow and a normalizing flow together toward a biased EBM for image generation. We use information geometry to understand the learning dynamics and show that the learned two-flow generator (i.e., CoopFlow) is a valid generative model, even though the learned EBM is biased.

**Joint training of EBM and normalizing flow.** There are other two works studying an EBM and a normalizing flow together as well. To avoid MCMC, Gao et al. (2020) propose to train an EBM using noise contrastive estimation (Gutmann & Hyvärinen, 2010), where the noise distribution is a normalizing flow. Nijkamp et al. (2020a) propose to learn an EBM as an exponential tilting of a pretrained normalizing flow, so that neural transport MCMC sampling in the latent space of the normalizing flow can mix well. Our paper proposes to train an EBM and a normalizing flow via short-run MCMC teaching. More specifically, we focus on short-run non-mixing MCMC and treat it as a valid flow-like model (i.e., short-run Langevin flow) that is guided by the EBM. Disregarding the biased EBM, the resulting valid generator is the combination of the short-run Langevin flow and the normalizing flow, where the latter serves as a rapid initializer of the former. The form of this two-flow generator shares a similar fashion with the Stochastic Normalizing Flow (Wu et al., 2020), which consists of a sequence of deterministic invertible transformations and stochastic sampling blocks.

## 3 TWO FLOW MODELS

### 3.1 LANGEVIN FLOW

**Energy-based model.** Let $x \in \mathbb{R}^D$ be the observed signal such as an image. An energy-based model defines an unnormalized probability distribution of $x$ as follows:

$$p_\theta(x) = \frac{1}{Z(\theta)} \exp[f_\theta(x)], \tag{1}$$

where $f_\theta : \mathbb{R}^D \to \mathbb{R}$ is the negative energy function and defined by a bottom-up neural network whose parameters are denoted by $\theta$. The normalizing constant or partition function $Z(\theta) = \int \exp[f_\theta(x)]dx$ is analytically intractable and difficult to compute due to high dimensionality of $x$.

**Maximum likelihood learning.** Suppose we observe unlabeled training examples $\{x_i, i = 1, ..., n\}$ from unknown data distribution $p_{\text{data}}(x)$, the energy-based model in Eq. (1) can be trained from $\{x_i\}$ by Markov chain Monte Carlo (MCMC)-based maximum likelihood estimation, in which MCMC samples are drawn from the model $p_\theta(x)$ to approximate the gradient of the log-likelihood function for updating the model parameters $\theta$. Specifically, the log-likelihood is $L(\theta) = \frac{1}{n} \sum_{i=1}^{n} \log p_\theta(x_i)$. For a large $n$, maximizing $L(\theta)$ is equivalent to minimizing the Kullback-Leibler (KL) divergence $\mathbb{D}_{\text{KL}}(p_{\text{data}} || p_\theta)$. The learning gradient is given by

$$L'(\theta) = \mathbb{E}_{p_{\text{data}}}[\nabla_\theta f_\theta(x)] - \mathbb{E}_{p_\theta}[\nabla_\theta f_\theta(x)] \approx \frac{1}{n} \sum_{i=1}^{n} \nabla_\theta f_\theta(x_i) - \frac{1}{n} \sum_{i=1}^{n} \nabla_\theta f_\theta(\tilde{x}_i), \tag{2}$$

where the expectations are approximated by averaging over the observed examples $\{x_i\}$ and the synthesized examples $\{\tilde{x}_i\}$ generated from the current model $p_\theta(x)$, respectively.

**Langevin dynamics as MCMC.** Generating synthesized examples from $p_\theta(x)$ can be accomplished with a gradient-based MCMC such as Langevin dynamics, which is applied as follows

$$x_t = x_{t-1} + \frac{\delta^2}{2}\nabla_x f_\theta(x_{t-1}) + \delta\epsilon_t; \quad x_0 \sim p_0(x), \quad \epsilon_t \sim \mathcal{N}(0, I_D), \quad t = 1, \cdots, T, \quad (3)$$

where $t$ indexes the Langevin time step, $\delta$ denotes the Langevin step size, $\epsilon_t$ is a Brownian motion that explores different modes, and $p_0(x)$ is a uniform distribution that initializes MCMC chains.

**Langevin flow.** As $T \to \infty$ and $\delta \to 0$, $x_T$ becomes an exact sample from $p_\theta(x)$ under some regularity conditions. However, it is impractical to run infinite steps with infinitesimal step size to generate fair examples from the target distribution. Additionally, convergence of MCMC chains in many cases is hopeless because $p_\theta(x)$ can be very complex and highly multi-modal, then the gradient-based Langevin dynamics has no way to escape from local modes, so that different MCMC chains with different starting points are unable to mix. Let $\tilde{p}_\theta(x)$ be the distribution of $x_T$, which is the resulting distribution of $x$ after $T$ steps of Langevin update starting from $x_0 \sim p_0(x)$. Due to the fixed $p_0(x)$, $T$ and $\delta$, the distribution $\tilde{p}_\theta(x)$ is well defined, which can be implicitly expressed by

$$\tilde{p}_\theta(x) = (\mathcal{K}_\theta p_0)(x) = \int p_0(z)\mathcal{K}_\theta(x|z)dz, \quad (4)$$

where $\mathcal{K}_\theta$ denotes the transition kernel of $T$ steps of Langevin dynamics that samples $p_\theta$. Generally, $\tilde{p}_\theta(x)$ is not necessarily equal to $p_\theta(x)$. $\tilde{p}_\theta(x)$ is dependent on $T$ and $s$, which are omitted in the notation for simplicity. The KL-divergence $\mathbb{D}_{\mathrm{KL}}(\tilde{p}_\theta(x)||p_\theta(x)) = -\mathrm{entropy}(\tilde{p}_\theta(x)) - \mathbb{E}_{\tilde{p}_\theta(x)}[f(x)] + \log Z$. The gradient ascent part of Langevin dynamics, $x_t = x_{t-1} + \delta^2/2\nabla_x f_\theta(x_{t-1})$, increases the negative energy $f_\theta(x)$ by moving $x$ to the local modes of $f_\theta(x)$, and the diffusion part $\delta\epsilon_t$ increases the entropy of $\tilde{p}_\theta(x)$ by jumping out of the local modes. According to the law of thermodynamics (Cover & Thomas, 2006), $\mathbb{D}_{\mathrm{KL}}(\tilde{p}_\theta(x)||p_\theta(x))$ decreases to zero monotonically as $T \to \infty$.

## 3.2 NORMALIZING FLOW

Let $z \in \mathbb{R}^D$ be the latent vector of the same dimensionality as $x$. A normalizing flow is of the form

$$x = g_\alpha(z); z \sim q_0(z), \quad (5)$$

where $q_0(z)$ is a known prior distribution such as Gaussian white noise distribution $\mathcal{N}(0, I_D)$, and $g_\alpha : \mathbb{R}^D \to \mathbb{R}^D$ is a mapping that consists of a sequence of $L$ invertible transformations, i.e., $g(z) = g_L \circ \cdots \circ g_2 \circ g_1(z)$, whose inversion $z = g_\alpha^{-1}(x)$ and log-determinants of the Jacobians can be computed in closed form. $\alpha$ are the parameters of $g_\alpha$. The mapping is used to transform a random vector $z$ that follows a simple distribution $q_0$ into a flexible distribution. Under the change-of-variables law, the resulting random vector $x = g_\alpha(z)$ has a probability density $q_\alpha(x) = q_0(g_\alpha^{-1}(x))|\det(\partial g_\alpha^{-1}(x)/\partial x)|$. Let $h_l = g_l(h_{l-1})$. The successive transformations between $x$ and $z$ can be expressed as a flow $z \xleftrightarrow{g_1} h_1 \xleftrightarrow{g_2} h_2 \cdots \xleftrightarrow{g_L} x$, where we define $z := h_0$ and $x := h_L$ for succinctness. Then the determinant becomes $|\det(\partial g_\alpha^{-1}(x)/\partial x)| = \prod_{l=1}^L |\det(\partial h_{l-1}/\partial h_l)|$. The log-likelihood of a datapoint $x$ can be easily computed by

$$\log q_\alpha(x) = \log q_0(z) + \sum_{l=1}^L \log \left|\det\left(\frac{\partial h_{l-1}}{\partial h_l}\right)\right|. \quad (6)$$

With some smart designs of the sequence of transformations $g_\alpha = \{g_l, l = 1, ..., L\}$, the log-determinant in Eq. (6) can be easily computed, then the normalizing flow $q_\alpha(x)$ can be trained by maximizing the exact data log-likelihood $L(\alpha) = \frac{1}{n}\sum_{i=1}^n \log q_\alpha(x_i)$ via gradient ascent algorithm.

## 4 COOPFLOW: COOPERATIVE TRAINING OF TWO FLOWS

### 4.1 COOPFLOW ALGORITHM

We have moved from trying to use a convergent Langevin dynamics to train a valid EBM. Instead, we accept the fact that the short-run non-convergent MCMC is inevitable and more affordable in practice, and we treats a non-convergent short-run Langevin flow as a generator and propose to jointly train it with a normalizing flow as a rapid initializer for more efficient generation. The resulting generator is called CoopFlow, which consists of both Langevin flow and normalizing flow.

Specifically, at each iteration, for $i = 1, ..., m$, we first generate $z_i \sim \mathcal{N}(0, I_D)$, and then transform $z_i$ by a normalizing flow to obtain $\hat{x}_i = g_\alpha(z_i)$. Next, starting from each $\hat{x}_i$, we run a Langevin flow (i.e., a finite number of Langevin steps toward an EBM $p_\theta(x)$) to obtain $\tilde{x}_i$. $\tilde{x}_i$ are considered synthesized examples that are generated by the CoopFlow model. We then update $\alpha$ of the normalizing flow by treating $\tilde{x}_i$ as training data, and update $\theta$ of the Langevin flow according to the learning gradient of the EBM, which is computed with the synthesized examples $\{\tilde{x}_i\}$ and the observed examples $\{x_i\}$. Algorithm 1 presents a description of the proposed CoopFlow algorithm. The advantage of this training scheme is that we only need to minimally modify the existing codes for the MLE training of EBM $p_\theta$ and normalizing flow $q_\alpha$. The probability density of the CoopFlow $\pi_{(\theta,\alpha)}(x)$ is well defined, which can be implicitly expressed by

$$\pi_{(\theta,\alpha)}(x) = (\mathcal{K}_\theta q_\alpha)(x) = \int q_\alpha(x') \mathcal{K}_\theta(x|x') dx'. \tag{7}$$

$\mathcal{K}_\theta$ is the transition kernel of the Langevin flow. If we increase the length $T$ of the Langevin flow, $\pi_{(\theta,\alpha)}$ will converge to the EBM $p_\theta(x)$. The network $f_\theta(x)$ in the Langevin flow is scalar valued and is of free form, whereas the network $g_\alpha(x)$ in the normalizing flow has high-dimensional output and is of a severely constrained form. Thus the Langevin flow can potentially provide a tighter fit to $p_{\text{data}}(x)$ than the normalizing flow. The Langevin flow may also be potentially more data efficient as it tends to have a smaller network than the normalizing flow. On the flip side, sampling from the Langevin flow requires multiple iterations, whereas the normalizing flow can synthesize examples via a direct mapping. It is thus desirable to train these two flows simultaneously, where the normalizing flow serves as an approximate sampler to amortize the iterative sampling of the Langevin flow. Meanwhile, the normalizing flow is updated by a temporal difference MCMC teaching provided by the Langevin flow, to further amortize the short-run Langevin flow.

---

**Algorithm 1** CoopFlow Algorithm

---

**Input**: (1) Observed images $\{x_i\}_i^n$; (2) Number of Langevin steps $T$; (3) Langevin step size $\delta$; (4) Learning rate $\eta_\theta$ for Langevin flow; (5) Learning rate $\eta_\alpha$ for normalizing flow; (6) batch size $m$.
**Output**: Parameters $\{\theta, \alpha\}$

 1: Randomly initialize $\theta$ and $\alpha$.
 2: **repeat**
 3:    Sample observed examples $\{x_i\}_i^m \sim p_{\text{data}}(x)$.
 4:    Sample noise examples $\{z_i\}_{i=1}^m \sim q_0(z)$,
 5:    Starting from $z_i$, generate $\hat{x}_i = g_\alpha(z_i)$ via normalizing flow.
 6:    Starting from $\hat{x}_i$, run a $T$-step Langevin flow to obtain $\tilde{x}_i$ by following Eq. (3).
 7:    Given $\{\tilde{x}_i\}$, update $\alpha$ by maximizing $\frac{1}{m}\sum_{i=1}^m \log q_\alpha(\tilde{x}_i)$ with Eq. (6).
 8:    Given $\{x_i\}$ and $\{\tilde{x}_i\}$, update $\theta$ by following the gradient $\nabla\theta$ in Eq. (2).
 9: **until** converged

---

### 4.2 Understanding the Learned Two Flows

**Convergence equations.** In the traditional contrastive divergence (CD) (Hinton, 2002) algorithm, MCMC chains are initialized with observed data so that the CD learning seeks to minimize $\mathbb{D}_{\text{KL}}(p_{\text{data}}(x)||p_\theta(x)) - \mathbb{D}_{\text{KL}}((\mathcal{K}_\theta p_{\text{data}})(x)||p_\theta(x))$, where $(\mathcal{K}_\theta p_{\text{data}})(x)$ denotes the marginal distribution obtained by running the Markov transition $\mathcal{K}_\theta$, which is specified by the Langevin flow, from the data distribution $p_{\text{data}}$. In the CoopFlow algorithm, the learning of the EBM (or the Langevin flow model) follows a modified contrastive divergence, where the initial distribution of the Langevin flow is modified to be a normalizing flow $q_\alpha$. Thus, at iteration $t$, the update of $\theta$ follows the gradient of $\mathbb{D}_{\text{KL}}(p_{\text{data}}||p_\theta) - \mathbb{D}_{\text{KL}}(\mathcal{K}_{\theta^{(t)}} q_{\alpha^{(t)}}||p_\theta)$ with respect to $\theta$. Compared to the traditional CD loss, the modified one replaces $p_{\text{data}}$ by $q_\alpha$ in the second KL divergence term. At iteration $t$, the update of the normalizing flow $q_\alpha$ follows the gradient of $\mathbb{D}_{\text{KL}}(\mathcal{K}_{\theta^{(t)}} q_{\alpha^{(t)}}||q_\alpha)$ with respect to $\alpha$. Let $(\theta^*, \alpha^*)$ be a fixed point the learning algorithm converges to, then we have the following convergence equations

$$\theta^* = \arg\min_\theta \mathbb{D}_{\text{KL}}(p_{\text{data}}||p_\theta) - \mathbb{D}_{\text{KL}}(\mathcal{K}_{\theta^*} q_{\alpha^*}||p_\theta), \tag{8}$$

$$\alpha^* = \arg\min_\alpha \mathbb{D}_{\text{KL}}(\mathcal{K}_{\theta^*} q_{\alpha^*}||q_\alpha). \tag{9}$$

**Ideal case analysis.** In the idealized scenario where the normalizing flow $q_\alpha$ has infinite capacity and the Langevin sampling can mix and converge to the sampled EBM, Eq. (9) means that $q_{\alpha^*}$

attempts to be the stationary distribution of $\mathcal{K}_{\theta^*}$, which is $p_{\theta^*}$ because $\mathcal{K}_{\theta^*}$ is the Markov transition kernel of a convergent MCMC sampled from $p_{\theta^*}$. That is, $\min_\alpha \mathbb{D}_{\text{KL}}(\mathcal{K}_{\theta^*}q_{\alpha^*}||q_\alpha) = 0$, thus $q_{\alpha^*} = p_{\theta^*}$. With the second term becoming 0 and vanishing, Eq. (8) degrades to $\min_\theta \mathbb{D}_{\text{KL}}(p_{\text{data}}||p_\theta)$ and thus $\theta^*$ is the maximum likelihood estimate. On the other hand, the normalizing flow $q_\alpha$ chases the EBM $p_\theta$ toward $p_{\text{data}}$, thus $\alpha^*$ is also a maximum likelihood estimate.

**Moment matching estimator.** In the practical scenario where the Langevin sampling is not mixing, the CoopFlow model $\pi_t = \mathcal{K}_{\theta^{(t)}}q_{\alpha^{(t)}}$ is an interpolation between the learned $q_{\alpha^{(t)}}$ and $p_{\theta^{(t)}}$, and it converges to $\pi^* = \mathcal{K}_{\theta^*}q_{\alpha^*}$, which is an interpolation between $q_{\alpha^*}$ and $p_{\theta^*}$. $\pi^*$ is the short-run Langevin flow starting from $q_{\alpha^*}$ towards EBM $p_{\theta^*}$. $\pi^*$ is a legitimate generator because $\mathbb{E}_{p_{\text{data}}}[\nabla_\theta f_{\theta^*}(x)] = \mathbb{E}_{\pi^*}[\nabla_\theta f_{\theta^*}(x)]$ at convergence. That is, $\pi^*$ leads to moment matching in the feature statistics $\nabla_\theta f_{\theta^*}(x)$. In other words, $\pi^*$ satisfies the above estimating equation.

**Understanding via information geometry**. Consider a simple EBM with $f_\theta(x) = \langle\theta, h(x)\rangle$, where $h(x)$ is the feature statistics. Since $\nabla_\theta f_\theta(x) = h(x)$, the MLE of the EBM $p_{\theta_{\text{MLE}}}$ is a moment matching estimator due to $\mathbb{E}_{p_{\text{data}}}[h(x)] = \mathbb{E}_{p_{\theta_{\text{MLE}}}}[h(x)]$. The CoopFlow $\pi^*$ also converges to a moment matching estimator, i.e., $\mathbb{E}_{p_{\text{data}}}[h(x)] = \mathbb{E}_{\pi^*}[h(x)]$. Figure 1 is an illustration of model distributions that correspond to different parameters at convergence. We first introduce three families of distributions: $\Omega = \{p : \mathbb{E}_p[h(x)] = \mathbb{E}_{p_{\text{data}}}[h(x)]\}$, $\Theta = \{p_\theta(x) = \exp(\langle\theta, h(x)\rangle)/Z(\theta), \forall\theta\}$, and $A = \{q_\alpha, \forall\alpha\}$, which are shown by the red, blue and green curves respectively in Figure 1.

$\Omega$ is the set of distributions that reproduce statistical property $h(x)$ of the data distribution. Obviously, $p_{\theta_{\text{MLE}}}$, $p_{\text{data}}$, and $\pi^* = \mathcal{K}_{\theta^*}q_{\alpha^*}$ are included in $\Omega$. $\Theta$ is the set of EBMs with different values of $\theta$, thus $p_{\theta_{\text{MLE}}}$ and $p_{\theta^*}$ belong to $\Theta$. Because of the short-run Langevin flow, $p_{\theta^*}$ is not a valid model that matches the data distribution in terms of $h(x)$, and thus $p_{\theta^*}$ is not in $\Omega$. $A$ is the set of normalizing flow models with different values of $\alpha$, thus $q_{\alpha^*}$ and $q_{\alpha_{\text{MLE}}}$ belong to $A$. The yellow line shows the MCMC trajectory. The solid segment of the yellow line, starting from $q_{\alpha^*}$ to $\mathcal{K}_{\theta^*}q_{\alpha^*}$, illustrates the short-run non-mixing MCMC that is initialized by the normalizing flow $q_{\alpha^*}$ in $A$ and arrives at $\pi^* = \mathcal{K}_{\theta^*}q_{\alpha^*}$ in $\Omega$. The dotted segment of the yellow line, starting from $\pi^* = \mathcal{K}_{\theta^*}q_{\alpha^*}$ in $\Omega$ to $p_{\theta^*}$ in $\Theta$, shows the potential long-run MCMC trajectory, which is not really realized because we stop short in MCMC. If we increase the number of steps of

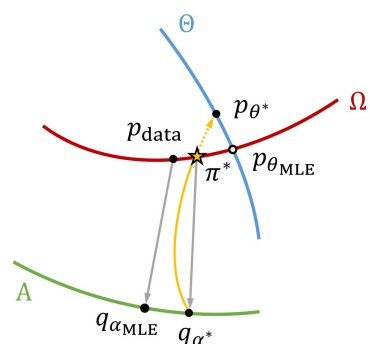

Figure 1: An illustration of convergence of the CoopFlow algorithm.

short-run Langevin flow, $\mathbb{D}_{\text{KL}}(\pi^*||p_{\theta^*})$ will be monotonically decreasing to 0. Though $\pi^*$ stops midway in the path toward $p_{\theta^*}$, $\pi^*$ is still a valid generator because it is in $\Omega$.

**Perturbation of MLE.** $p_{\theta_{\text{MLE}}}$ is the intersection between $\Theta$ and $\Omega$. It is the projection of $p_{\text{data}}$ onto $\Theta$ because $\theta_{\text{MLE}} = \arg\min_\theta \mathbb{D}_{\text{KL}}(p_{\text{data}}||p_\theta)$. $q_{\alpha_{\text{MLE}}}$ is also a projection of $p_{\text{data}}$ onto $A$ because $\alpha_{\text{MLE}} = \arg\min_\alpha \mathbb{D}_{\text{KL}}(p_{\text{data}}||q_\alpha)$. Generally, $q_{\alpha_{\text{MLE}}}$ is far from $p_{\text{data}}$ due to its restricted form network, whereas $p_{\theta_{\text{MLE}}}$ is very close to $p_{\text{data}}$ due to its scalar-valued free form network. Because the short-run MCMC is not mixing, $\theta^* \neq \theta_{\text{MLE}}$ and $\alpha^* \neq \alpha_{\text{MLE}}$. $\theta^*$ and $\alpha^*$ are perturbations of $\theta_{\text{MLE}}$ and $\alpha_{\text{MLE}}$, respectively. $p_{\theta_{\text{MLE}}}$ shown by an empty dot is not attainable unfortunately. We want to point out that, as $T$ goes to infinity, $p_{\theta^*} = p_{\theta_{\text{MLE}}}$, and $q_{\alpha^*} = q_{\alpha_{\text{MLE}}}$. Note that $\Omega$, $\Theta$ and $A$ are high-dimensional manifolds instead of 1D curves as depicted in Figure 1, and $\pi^*$ may be farther away from $p_{\text{data}}$ than $p_{\theta_{\text{MLE}}}$ is. During learning, $q_{\alpha^{(t+1)}}$ is the projection of $\mathcal{K}_{\theta^{(t)}}q_{\alpha^{(t)}}$ on $A$. At convergence, $q_{\alpha^*}$ is the projection of $\pi^* = \mathcal{K}_{\theta^*}q_{\alpha^*}$ on $A$. There is an infinite looping between $q_{\alpha^*}$ and $\pi^* = \mathcal{K}_{\theta^*}q_{\alpha^*}$ at convergence of the CoopFlow algorithm, i.e., $\pi^*$ lifts $q_{\alpha^*}$ off the ground $A$, and the projection drops $\pi^*$ back to $q_{\alpha^*}$. Although $p_{\theta^*}$ and $q_{\alpha^*}$ are biased, they are not wrong. Many useful models and algorithms, e.g., variational autoencoder and contrastive divergence are also biased to MLE. Their learning objectives also follow perturbation of MLE.

## 5 EXPERIMENTS

We showcase experiment results on various tasks. We start from a toy example to illustrate the basic idea of the CoopFlow in Section 5.1. We show image generation results in Section 5.2. Section 5.3 demonstrates the learned CoopFlow is useful for image reconstruction and inpainting, while Section 5.4 shows that the learned latent space is meaningful so that it can be used for interpolation.

## 5.1 TOY EXAMPLE STUDY

We first demonstrate our idea using a two-dimensional toy example where data lie on a spiral. We train 3 CoopFlow models with different lengths of Langevin flows. As shown in Figure 2, the rightmost box shows the results obtained with 10,000 Langevin steps. We can see that both the normalizing flow $q_\alpha$ and the EBM $p_\theta$ can fit the ground truth density, which is displayed in the red box, perfectly. This validates that, with a sufficiently long Langevin flow, the CoopFlow algorithm can learn both a valid $q_\alpha$ and a valid $p_\theta$. The leftmost green box represents the model trained with 100 Langevin steps. This is a typical non-convergent short-run MCMC setting. In this case, neither $q_\alpha$ nor $p_\theta$ is valid, but their cooperation is. The short-run Langevin dynamics toward the EBM actually works as an flow-like generator that modifies the initial proposal by the normalizing flow. We can see that the samples from $q_\alpha$ are not perfect, but after the modification, the samples from $\pi_{(\theta,\alpha)}$ fit the ground truth distribution very well. The third box shows the results obtained with 500 Langevin steps. This is still a short-run setting, even though it uses more Langevin steps. $p_\theta$ becomes better than that with 100 steps, but it is still invalid. With an increased number of Langevin steps, samples from both $q_\alpha$ and $\pi_{(\theta,\alpha)}$ are improved and comparable to those in the long-run setting with 10,000 steps. The results verify that the CoopFlow might learn a biased BEM and a biased normalizing flow if the Langevin flow is non-convergent. However, the non-convergent Langevin flow together with the biased normalizing flow can still form a valid generator that synthesizes valid examples.

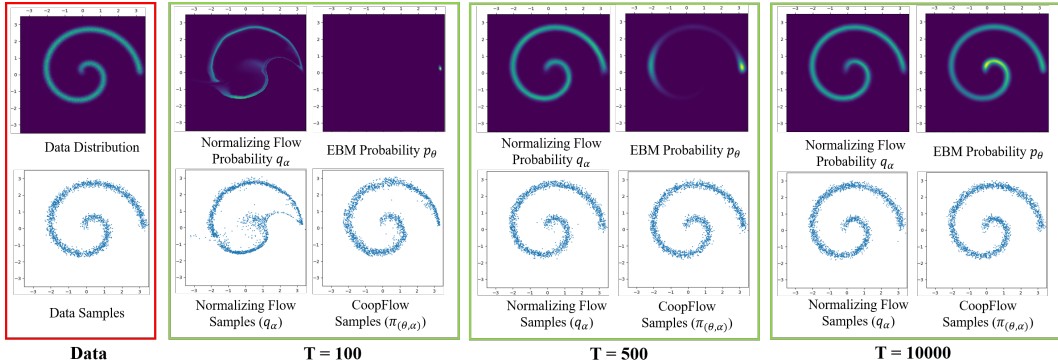

Figure 2: Learning CoopFlows on two-dimensional data. The ground truth data distribution is shown in the red box, and the models trained with different Langevin steps are in the three green boxes. In each green box, the first row shows the learned distributions of the normalizing flow and the EBM, and the second row shows the samples from the learned normalizing flow and the learned CoopFlow.

## 5.2 IMAGE GENERATION

We test our model on 3 image datasets for image synthesis. (i) CIFAR-10 (Krizhevsky & Hinton, 2009) is a dataset containing 50k training images and 10k testing images in 10 classes; (ii) SVHN (Netzer et al., 2011) is a dataset containing over 70k training images and over 20k testing images of house numbers; (iii) CelebA (Liu et al., 2015) is a celebrity facial dataset containing over 200k images. We downsample all the images to the resolution of $32 \times 32$. For our model, we show results of three different settings. *CoopFlow(T=30)* denotes the setting where we train a normalizing flow and a Langevin flow together from scratch and use 30 Langevin steps. *CoopFlow(T=200)* denotes the setting that we increase the number of Langevin steps to 200. In the *CoopFlow(Pre)* setting, we first pretrain a normalizing flow from observed data, and then train the CoopFlow with the parameters of the normalizing flow being initialized by the pretrained one. We use a 30-step-long Langevin flow in this setting. For all the three settings, we slightly increase the Langevin step size at the testing stage for better performance. We show both qualitative results in Figure 3 and quantitative results in Table 1. To calculate FID Heusel et al. (2017) scores, we generate 50,000 samples on each dataset. Our models outperform most of the baseline algorithms. We get lower FID scores comparing to the single normalizing flow models and the prior works that jointly train a normalizing flow with an EBM, e.g., Gao et al. (2020); Nijkamp et al. (2020a). We also achieve comparable results with the state-of-the-art EBMs. We can see using more Langevin steps or a pretrained normalizing flow can help improve the performance of the CoopFlow. The former enhances the expressive power, while the latter stabilizes the training. More experimental details and results can be found in Appendix.

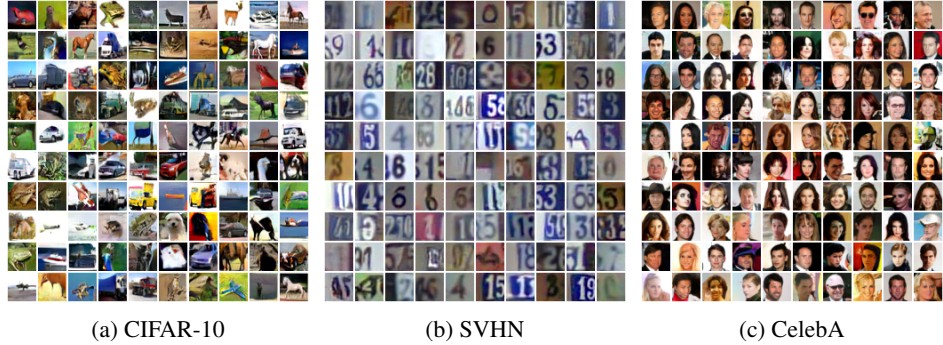

| (a) CIFAR-10 | (b) SVHN | (c) CelebA |

Figure 3: Generated examples ($32 \times 32$ pixels) by the CoopFlow models trained on the CIFAR-10, SVHN and CelebA datasets respectively. Samples are obtained from the setting of *CoopFlow(pre)*.

|  | Models | FID ↓ |
|---|---|---|
| VAE | VAE (Kingma & Welling, 2014) | 78.41 |
| Autoregressive | PixelCNN (Salimans et al., 2017) | 65.93 |
|  | PixelIQN (Ostrovski et al., 2018) | 49.46 |
| GAN | WGAN-GP(Gulrajani et al., 2017) | 36.40 |
|  | SN-GAN (Miyato et al., 2018) | 21.70 |
|  | StyleGAN2-ADA (Karras et al., 2020) | 2.92 |
| Score-Based | NCSN (Song & Ermon, 2019) | 25.32 |
|  | NCSN-v2 (Song & Ermon, 2020) | 31.75 |
|  | NCSN++ (Song et al., 2021) | 2.20 |
| Flow | Glow (Kingma & Dhariwal, 2018) | 45.99 |
|  | Residual Flow (Chen et al., 2019) | 46.37 |
| EBM | LP-EBM (Pang et al., 2020) | 70.15 |
|  | EBM-SR (Nijkamp et al., 2019) | 44.50 |
|  | EBM-IG (Du & Mordatch, 2019) | 38.20 |
|  | CoopVAEBM (Xie et al., 2021b) | 36.20 |
|  | CoopNets (Xie et al., 2020a) | 33.61 |
|  | Divergence Triangle (Han et al., 2020) | 30.10 |
|  | VARA (Grathwohl et al., 2021) | 27.50 |
|  | EBM-CD (Du et al., 2021) | 25.10 |
|  | GEBM (Arbel et al., 2021) | 19.31 |
|  | CF-EBM Zhao et al. (2021) | 16.71 |
|  | VAEBM (Xiao et al., 2021) | 12.16 |
|  | EBM-Diffusion (Gao et al., 2021) | 9.60 |
| Flow+EBM | NT-EBM (Nijkamp et al., 2020a) | 78.12 |
|  | EBM-FCE (Gao et al., 2020) | 37.30 |
| Ours | CoopFlow(T=30) | 21.16 |
|  | CoopFlow(T=200) | 18.89 |
|  | CoopFlow(Pre) | 15.80 |

(a) CIFAR-10

| Models | FID ↓ |
|---|---|
| ABP (Han et al., 2017) | 49.71 |
| ABP-SRI (Nijkamp et al., 2020b) | 35.23 |
| ABP-OT (An et al., 2021) | 19.48 |
| VAE (Kingma & Welling, 2014) | 46.78 |
| 2sVAE (Dai & Wipf, 2019) | 42.81 |
| RAE (Ghosh et al., 2020) | 40.02 |
| Glow (Kingma & Dhariwal, 2018) | 41.70 |
| DCGAN (Radford et al., 2016) | 21.40 |
| NT-EBM (Nijkamp et al., 2020a) | 48.01 |
| LP-EBM (Pang et al., 2020) | 29.44 |
| EBM-FCE (Gao et al., 2020) | 20.19 |
| CoopFlow(T=30) | 18.11 |
| CoopFlow(T=200) | 16.97 |
| CoopFlow(Pre) | 15.32 |

(b) SVHN

| Models | FID ↓ |
|---|---|
| ABP (Han et al., 2017) | 51.50 |
| ABP-SRI (Nijkamp et al., 2020b) | 36.84 |
| VAE (Kingma & Welling, 2014) | 38.76 |
| Glow (Kingma & Dhariwal, 2018) | 23.32 |
| DCGAN (Radford et al., 2016) | 12.50 |
| EBM-FCE (Gao et al., 2020) | 12.21 |
| GEBM (Arbel et al., 2021) | 5.21 |
| CoopFlow(T=30) | 6.44 |
| CoopFlow(T=200) | 4.90 |
| CoopFlow(Pre) | 4.15 |

(c) CelebA

Table 1: FID scores on CIFAR-10, SVHN and CelebA datasets. Images are resized to $32 \times 32$ pixels.

## 5.3 Image Reconstruction and Inpainting

We show that the learned CoopFlow model is able to reconstruct observed images. We may consider the CoopFlow model $\pi_{(\theta,\alpha)}(x)$ a latent variable generative model: $z \sim q_0(z); \hat{x} = g_\alpha(z); x = F_\theta(\hat{x}, e)$, where $z$ denotes the latent variables, $e$ denotes all the injected noises in the Langevin flow, and $F_\theta$ denotes the mapping realized by a $T$-step Langevin flow that is actually a $T$-layer noise-injected residual network. Since the Langevin flow is not mixing, $x$ is dependent on $\hat{x}$ in the Langevin flow, thus also dependent on $z$. The CoopFlow is a generator $x = F_\theta(g_\alpha(z), e)$, so we can reconstruct any $x$ by inferring the corresponding latent variables $z$ using gradient descent on $L(z) = ||x - F_\theta(g_\alpha(z), e)||^2$, with $z$ being initialized by $q_0$. However, $g$ is an invertible transformation, so we can infer $z$ by an efficient way, i.e., we first find $\hat{x}$ by gradient descent on $L(\hat{x}) = ||x - F_\theta(\hat{x}, e)||^2$, with $\hat{x}$ being initialized by $\hat{x}_0 = g_\alpha(z)$ where $z \sim p_0(z)$ and $e$ being set to be 0, and then use $z = g_\alpha^{-1}(\hat{x})$ to get the latent variables. These two methods are equivalent, but the latter one is computationally efficient, since computing the gradient on the whole two-flow generator $F_\theta(g_\alpha(z), e)$ is difficult and time-consuming. Let $\hat{x}^* = \arg\min_{\hat{x}} L(\hat{x})$. The reconstruction is given by $F_\theta(\hat{x}^*)$. The optimization is done using 200 steps of gradient descent over $\hat{x}$.

The reconstruction results are shown in Figure 4. We use the CIFAR-10 testing set. The right column displays the original images $x$ that need to be reconstructed. The left and the middle columns display $\hat{x}^*$ and $F_\theta(\hat{x}^*)$, respectively. Our model can successfully reconstruct the observed images, verifying that the CoopFlow with a non-mixing MCMC is indeed a valid latent variable model.

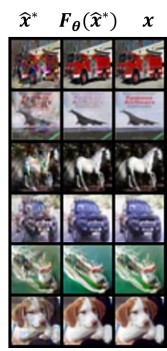

$\hat{x}^*$ $\quad F_\theta(\hat{x}^*)$ $\quad x$

Figure 4: Image reconstruction on the CIFAR-10.

We further show that our model is also capable of doing image inpainting. Similar to image reconstruction, given a masked observation $x_{\text{mask}}$ along with a binary matrix $M$ indicating the positions of the unmasked pixels, we optimize $\hat{x}$ to minimize the reconstruction error between $F_\theta(\hat{x})$ and $x_{\text{mask}}$ in the unmasked area, i.e., $L(\hat{x}) = ||M \odot (x_{\text{mask}} - F_\theta(\hat{x}))||^2$, where $\odot$ is the element-wise multiplication operator. $\hat{x}$ is still initialized by the normalizing flow. We do experiments on the CelebA training set. In Figure 5, each row shows one example of inpainting with a different initialization of $\hat{x}$ provided by the normalizing flow and the first 17 columns display the inpainting results at different optimization iterations. The last two columns show the masked images and the original images respectively. We can see that our model can reconstruct the unmasked areas faithfully and simultaneously fill in the blank areas of the input images. With different initializations, our model can inpaint diversified and meaningful patterns.

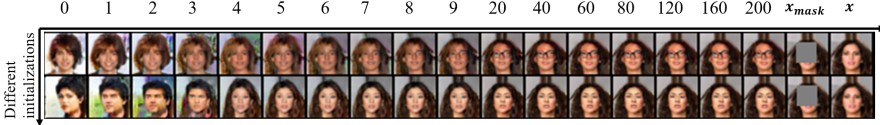

Figure 5: Image inpainting on the CelebA dataset ($32 \times 32$ pixels). Each row represents one different initialization. The last two columns display the masked and original images respectively. From the 1st column to the 17th column, we show the inpainted images at different optimization iterations.

## 5.4 INTERPOLATION IN THE LATENT SPACE

The CoopFlow model is capable of doing interpolation in the latent space $z$. Given an image $x$, we first find its corresponding $\hat{x}^*$ using the reconstruction method described in Section 5.3. We then infer $z$ by the inversion of the normalizing flow $z^* = g_\alpha^{-1}(\hat{x}^*)$. Figure 6 shows two examples of interpolation between two latent vectors inferred from observed images. For each row, the images at the two ends are observed. Each image in the middle is obtained by first interpolating the latent vectors of the two end images, and then generating the image using the CoopFlow generator, This experiment shows that the CoopFlow can learn a smooth latent space that traces the data manifold.

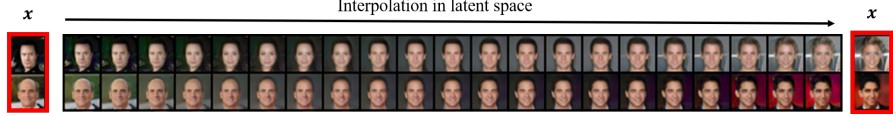

Figure 6: Image interpolation results on the CelebA dataset ($32 \times 32$ pixels). The leftmost and rightmost columns display the images we observed. The columns in the middle represent the interploation results between the inferred latent vectors of the two end observed images.

## 6 CONCLUSION

This paper studies an interesting problem of learning two types of deep flow models in the context of energy-based framework for image representation and generation. One is the normalizing flow that generates synthesized examples by transforming Gaussian noise examples through a sequence of invertible transformations, while the other is the Langevin flow that generates synthesized examples by running a non-mixing, non-convergent short-run MCMC toward an EBM. We propose the CoopFlow algorithm to train the short-run Langevin flow model jointly with the normalizing flow serving as a rapid initializer in a cooperative manner. The experiments show that the CoopFlow is a valid generative model that can be useful for image generation, reconstruction, and interpolation.

ACKNOWLEDGMENTS

The authors sincerely thank Dr. Ying Nian Wu for the helpful discussion on the information geometry part and thank Dr. Yifei Xu for his assistance with some extra experiments during the rebuttal. The authors would also like to thank the anonymous reviewers of ICLR'22 program committee for providing constructive comments and suggestions to improve the work.

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

# A APPENDIX

## A.1 NETWORK ARCHITECTURE IN THE COOPFLOW

We use the same network architecture for all the experiments. For the normalizing flow $g_\alpha(z)$ in our CoopFlow framework, we use the Flow++ (Ho et al., 2019) network architecture that was originally designed for the CIFAR-10 ($32 \times 32$ pixels) dataset in Ho et al. (2019). As to the EBM in our CoopFlow, we use the architecture shown in Table 2 to design the negative energy function $f_\theta(x)$.

| |
|---|
| $3 \times 3$ Conv2d, *str*=1, *pad*=1, *ch*=128; Swish |
| Residual Block, *ch*=256 |
| Residual Block, *ch*=512 |
| Residual Block, *ch*=1,024 |
| $3 \times 3$ Conv2d, *str*=4, *pad*=0, *ch*=100; Swish |
| Sum over channel dimension |

(a) EBM architecture

| |
|---|
| $3 \times 3$ Conv2d, *str*=1, *pad*=1; Swish |
| $3 \times 3$ Conv2d, *str*=1, *pad*=1; Swish |
| + $3 \times 3$ Conv2d, *str*=1, *pad*=1; Swish (input) |
| $2 \times 2$ Average pooling |

(b) Residual Block architecture

Table 2: Network architecture of the EBM in the CoopFlow (*str*: stride, *pad*: padding, *ch*: channel).

## A.2 EXPERIMENTAL DETAILS

We have three different settings for the CoopFlow model. In the *CoopFlow(T=30)* setting and *CoopFlow(T=200)* setting, we train both the normalizing flow and the Langevin flow from scratch. The difference between them are only the number of the Langevin steps. The *CoopFlow(T=200)* uses a longer Langevin flow than the *CoopFlow(T=30)*. We follow Ho et al. (2019) and use the data-dependent parameter initialization method (Salimans & Kingma, 2016) for our normalizing flow in both settings *CoopFlow(T=30)* and *CoopFlow(T=200)*. On the other hand, as to the *CoopFlow(Pre)* setting, we first pretrain a normalizing flow on training examples, and then train a 30-step Langevin flow, whose parameters are initialized randomly, together with the pretrained normalizing flow by following Algorithm 1. The cooperation between the pretrained normalizing flow and the untrained Langevin flow would be difficult and unstable because the Langevin flow is not knowledgeable at all to teach the normalizing flow. To stabilize the cooperative training and make a smooth transition for the normalizing flow, we include a warm-up phase in the CoopFlow algorithm. During this phase, instead of updating both the normalizing flow and the Langevin flow, we fix the parameters of the pretrained normalizing flow and only update the parameters of the Langevin flow. After a certain number of learning epochs, the Langevin flow may get used to the normalizing flow initialization, and learn to cooperate with it. Then we begin to update both two flows as described in Algorithm 1. This strategy is effective in preventing the Langevin flow from generating bad synthesized examples at the beginning of the CoopFlow algorithm to ruin the pretrained normalizing flow.

We use the Adam optimizer (Kingma & Ba, 2015) for training. We set learning rates $\eta_\alpha = 0.0001$ and $\eta_\theta = 0.0001$ for the normalizing flow and the Langevin flow, respectively. We use $\beta_1 = 0.9$ and $\beta_2 = 0.999$ for the normalizing flow and $\beta_1 = 0.5$ and $\beta_2 = 0.5$ for the Langevin flow. In the Adam optimizer, $\beta_1$ is the exponential decay rate for the first moment estimates, and $\beta_2$ is the exponential decay rate for the second-moment estimates. We adopt random horizontal flip as data augmentation only for the CIFAR-10 dataset. We remove the noise term in each Langevin update by following Zhao et al. (2021). We also propose an alternative strategy to gradually decay the effect of the noise term in Section A.11. The batch sizes for *CoopFlow(T=30)*, *CoopFlow(T=200)*, and *CoopFlow(Pre)* are 28, 32 and 28. The values of other hyperparameters can be found in Table 3.

## A.3 ANALYSIS OF HYPERPARAMETERS OF LANGEVIN FLOW

We investigate the influence of the Langevin step size $\delta$ and the number of Langevin steps $T$ on the CIFAR-10 dataset using the *CoopFlow(Pre)* setting. We first fix the Langevin step size to be 0.03 and vary the number of Langevin steps from 10 to 50. The results are shown in Table 4. On the other hand, we show the influence of the Langevin step size in Table 5, where we fix the number

| Model | Dataset | # of epochs to pretrain Normal. Flow | # of warm-up epochs for Lang. Flow | # of epochs for CoopFlow | # of Langevin steps | Langevin step size (train) | Langevin step size (test) |
|-------|---------|------|------|------|------|------|------|
| CoopFlow (T=30) | CIFAR-10 | 0 | 0 | 100 | 30 | 0.03 | 0.04 |
|  | SVHN | 0 | 0 | 100 | 30 | 0.03 | 0.035 |
|  | CelebA | 0 | 0 | 100 | 30 | 0.03 | 0.035 |
| CoopFlow (T=200) | CIFAR-10 | 0 | 0 | 100 | 200 | 0.01 | 0.012 |
|  | SVHN | 0 | 0 | 100 | 200 | 0.011 | 0.0125 |
|  | CelebA | 0 | 0 | 100 | 200 | 0.011 | 0.013 |
| CoopFlow (Pre) | CIFAR-10 | 300 | 25 | 75 | 30 | 0.03 | 0.04 |
|  | SVHN | 200 | 10 | 90 | 30 | 0.03 | 0.035 |
|  | CelebA | 80 | 10 | 90 | 30 | 0.03 | 0.035 |

Table 3: Hyperparameter setting in our experiments.

of Langevin steps to be 30 and vary the Langevin step size used in training. When synthesizing examples from the learned models in testing, we slightly increase the Langevin step size by a ratio of $4/3$ for better performance. We can see that our choices of 30 as the number of Langevin steps and 0.03 as the Langevin step size are reasonable. Increasing the number of Langevin steps can improve the performance in terms of FID, but also be computationally expensive. The choice of $T = 30$ is a trade-off between the synthesis performance and the computation efficiency.

| # of Langevin steps | 10 | 20 | 30 | 40 | 50 |
|---------------------|-----|-----|-----|-----|-----|
| FID $\downarrow$ | 16.46 | 15.20 | 15.80 | 16.80 | 15.64 |

Table 4: FID scores over the numbers of Langevin steps of *CoopFlow(Pre)* on the CIFAR-10 dataset.

| Langevin step size (train) | 0.01 | 0.02 | 0.03 | 0.04 | 0.05 | 0.10 |
|----------------------------|------|------|------|------|------|------|
| Langevin step size (test) | 0.013 | 0.026 | 0.04 | 0.053 | 0.067 | 0.13 |
| FID $\downarrow$ | 15.99 | 16.32 | 15.80 | 16.52 | 18.17 | 19.82 |

Table 5: FID scores of *CoopFlow(Pre)* on the CIFAR-10 dataset under different Langevin step sizes.

## A.4 ABLATION STUDY

To show the effect of the cooperative training, we compare a CoopFlow model with an individual normalizing flow and an individual Langevin flow. For fair comparison, the normalizing flow component in the CoopFlow has the same network architecture as that in the individual normalizing flow, while the Langevin flow component in the CoopFlow also uses the same network architecture as that in the individual Langevin flow. We train the individual normalizing flow by following Ho et al. (2019) and train the individual Langevin flow by following Nijkamp et al. (2019). All three models are trained on the CIFAR-10 dataset. We present a comparison of these three models in terms of FID in Table 6, and also show generated samples in Figure 7. From Table 6, we can see that the CoopFlow model outperforms both the normalizing flow and the Langevin flow by a large margin, which verifies the effectiveness of the proposed CoopFlow algorithm.

| Model | Normalizing flow | Langevin flow | CoopFlow |
|-------|------------------|---------------|----------|
| FID $\downarrow$ | 92.10 | 49.51 | 21.16 |

Table 6: A FID comparison among the normalizing flow, the Langevin flow and the CoopFlow.

## A.5 MORE IMAGE GENERATION RESULTS

In Section 5.2, we have shown generated examples from the *CoopFlow(Pre)* model. In this section, we show examples generated by the *CoopFlow(T=30)* model in Figure 8 and examples generated by the *CoopFlow(T=200)* in Figure 9. We can see that all the generated examples are meaningful.

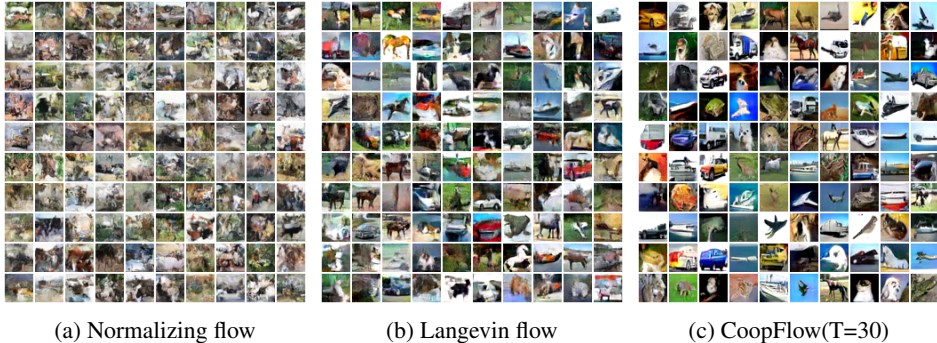

(a) Normalizing flow        (b) Langevin flow        (c) CoopFlow(T=30)

Figure 7: Generated examples by (a) the individual normalizing flow, (b) the individual Langevin flow, and (c) the CoopFlow, which are trained on the CIFAR-10 dataset.

On the other hand, we show the examples generated from the normalizing flow component of the CoopFlow model in Figure 10. By comparing the synthesized images shown in Figure 8 and those in Figure 10, we can see that there is a gap between the normalizing flow and the CoopFlow. The samples from the normalizing flow look blurred but become sharp and clear after the Langevin flow revision. This supports our claim in Section 4.2.

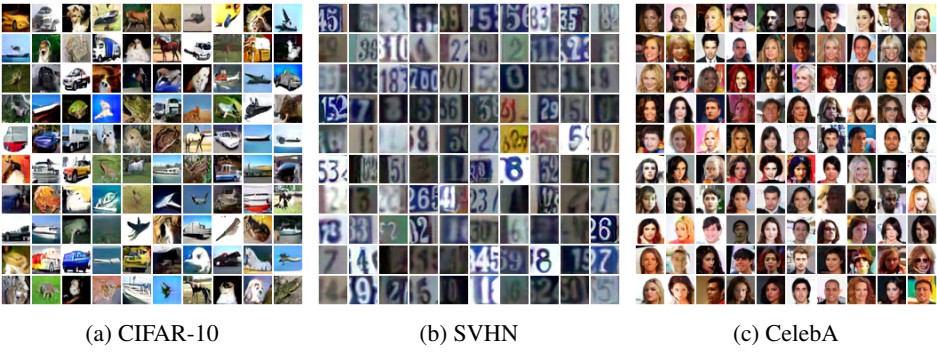

(a) CIFAR-10        (b) SVHN        (c) CelebA

Figure 8: Generated examples ($32 \times 32$ pixels) by the CoopFlow models trained on the CIFAR-10, SVHN and CelebA datasets respectively. Samples are obtained from the *CoopFlow(T=30)* setting.

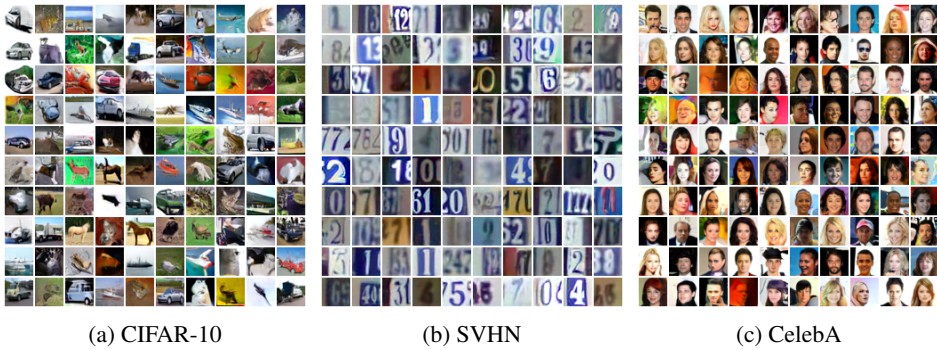

(a) CIFAR-10        (b) SVHN        (c) CelebA

Figure 9: Generated examples ($32 \times 32$ pixels) by the CoopFlow models trained on the CIFAR-10, SVHN and CelebA datasets respectively. Samples are obtained from the *CoopFlow(T=200)* setting.

## A.6  MORE IMAGE INPAINTING RESULTS

We show more image inpainting results in Figure 11. Each panel corresponds to one inpainting task. Each row in each panel represents one inpainting result with a different initialization. In each panel, the last two columns show the masked and the ground truth images respectively, and images from the 1st column to the 20th column are the inpainted images at different optimization iterations.

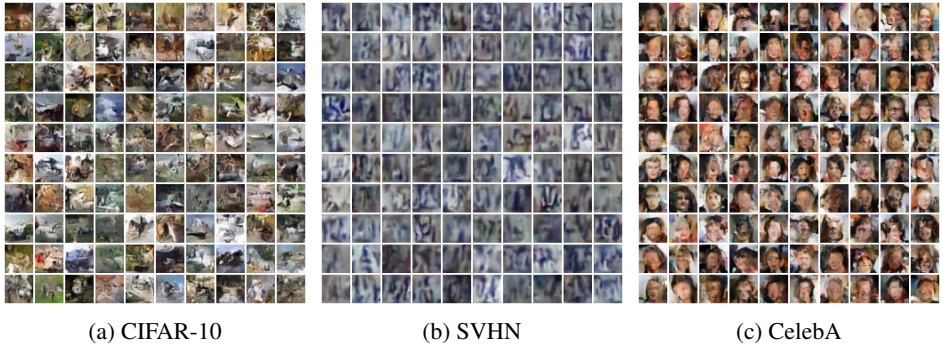

(a) CIFAR-10          (b) SVHN          (c) CelebA

Figure 10: Generated examples by the normalizing flow components in the CoopFlow models trained on the CIFAR-10, SVHN, and CelebA datasets respectively. The CoopFlow models are trained in the *CoopFlow(T=30)* setting.

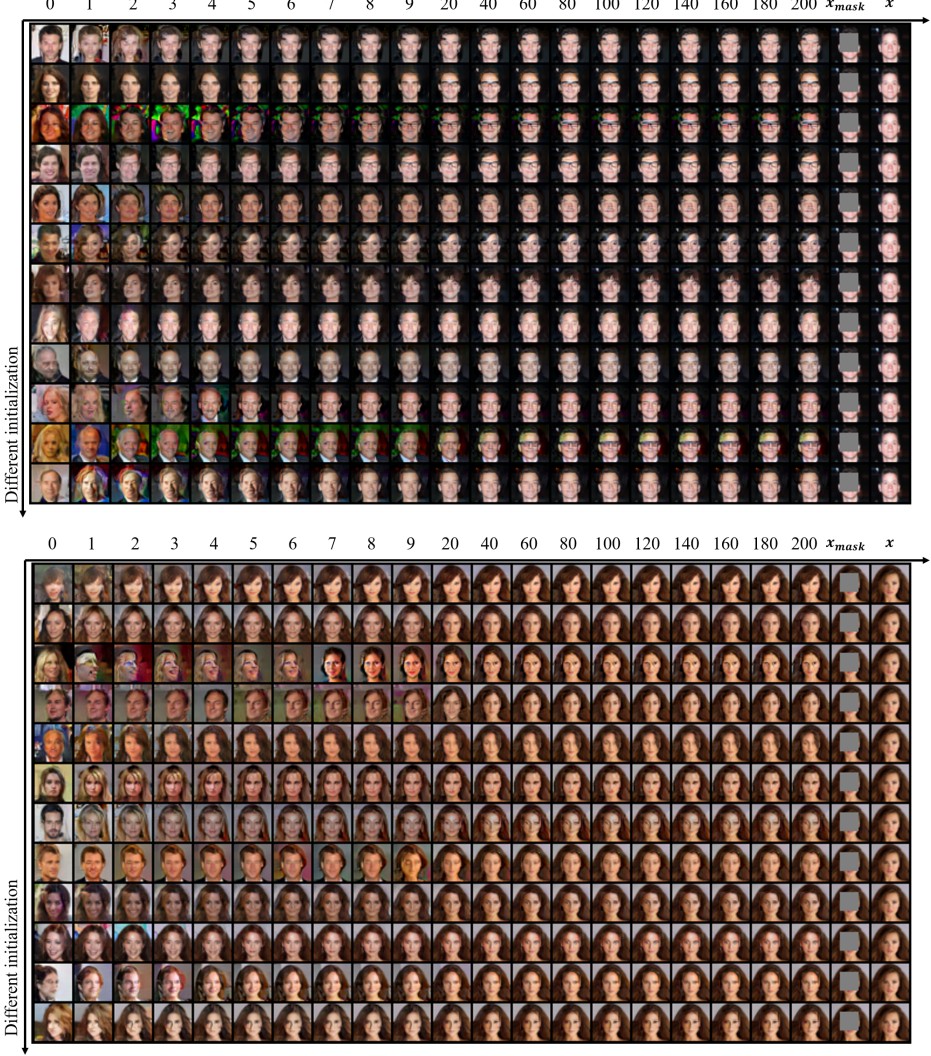

Figure 11: More results on image inpainting.

### A.7 FID Curve over Training Epochs

Figure 12 shows the FID trends during the training of the CoopFlow models on the CIFAR-10 dataset. Each curve represents the FID scores over training epochs. We train the CoopFlow models using the settings of *CoopFlow(T=30)* and *CoopFlow(pre)*. For each of them, we can observe that, as the cooperative learning proceeds, the FID keeps decreasing and converges to a low value. This is an empirical evidence to show that the proposed CoopFlow algorithm is a convergent algorithm.

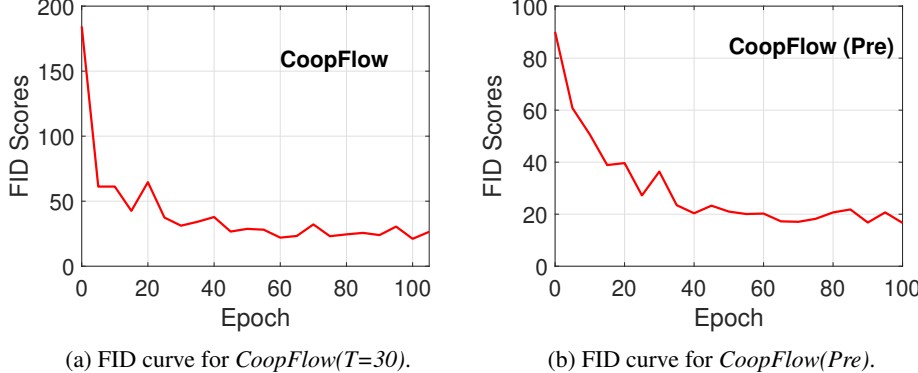

(a) FID curve for *CoopFlow(T=30)*.            (b) FID curve for *CoopFlow(Pre)*.

Figure 12: FID curves on the CIFAR-10 dataset. The FID score is reported every 5 epochs.

### A.8 Quantitative Results For Image Reconstruction

We provide additional quantitative results for the image reconstruction experiment in Section 5.3. Following Nijkamp et al. (2019), we calculate the per-pixel MSE on 1,000 examples in the testing set of the CIFAR-10 dataset. We use a 200-step gradient descent to minimize the reconstruction loss. We plot the reconstruction error curve showing the MSEs over iterations in Figure 13 and report the final per-pixel MSE in Table 7. For a baseline, we train an EBM using a 100-step short-run MCMC and the resulting model is the short-run Langevin flow. We then apply it to the reconstruction of the same 1,000 images by following Nijkamp et al. (2019). The baseline EBM has the same network architecture as that of the EBM component in our CoopFlow model for fair comparison. The experiment results show that the CoopFlow works better than the individual short-run Langevin flow in this image reconstruction task.

| Model | MSE |
|---|---|
| Langevin flow / EBM with short-run MCMC | 0.1083 |
| CoopFlow (ours) | 0.0254 |

Table 7: Reconstruction error (MSE per pixel).

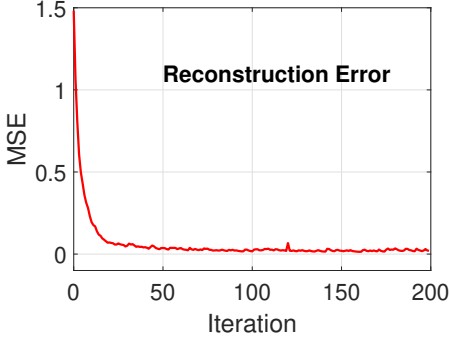

Figure 13: Reconstruction errors (MSE per pixel) over iterations.

## A.9 MODEL COMPLEXITY

In Table 8, we present a comparison of different models in terms of model size and FID score. Here we mainly compare those models that have a normalizing flow component, e.g., EBM-FCE, NT-EBM, GLOW, Flow++, as well as an EBM jointly trained with a VAE generator, e.g., VAEBM. We can see the CoopFlow model has a good balance between model complexity and performance. It is noteworthy that both the CoopFlow and the EBM-FCE consist of an EBM and a normalizing flow, and also have similar model sizes, but the CoopFlow achieves a much lower FID than the EBM-FCE. Note that the Flow++ baseline uses the same structure as the one in our CoopFlow. By comparing the Flow++ and the CoopFlow, we can find that recruiting an extra Langevin Flow can help improve the performance of the normalizing flow in terms of FID. On the other hand, although the VAEBM model achieves a better FID than ours, but it relies on a much larger pretrained NVAE model (Vahdat & Kautz, 2020) that significantly increases the model complexity.

| Model | # of Parameters | FID $\downarrow$ |
|---|---|---|
| NT-EBM (Nijkamp et al., 2020a) | 23.8M | 78.12 |
| GLOW (Kingma & Dhariwal, 2018) | 44.2M | 45.99 |
| EBM-FCE (Gao et al., 2020) | 44.9M | 37.30 |
| Flow++ (Ho et al., 2019) | 28.8M | 92.10 |
| VAEBM (Xiao et al., 2021) | 135.1M | 12.16 |
| CoopFlow(T=30) (ours) | 45.9M | 21.16 |
| CoopFlow(T=200) (ours) | 45.9M | 18.89 |
| CoopFlow(pre) (ours) | 45.9M | 15.80 |

Table 8: A comparison of model sizes and FID scores for different models. FID scores are reported on the CIFAR-10 dataset.

## A.10 COMPARISON WITH MODELS USING SHORT-RUN MCMC

Our method is relevant to the short-run MCMC. In this section, we compare the CoopFlow model with other models that use a short-run MCMC as a flow-like generator. The baselines include (1) the single EBM with short-run MCMC starting from the noise distribution (Nijkamp et al., 2019), and (ii) cooperative training of an EBM and a generic generator (Xie et al., 2020a). In Table 9, we report the FID scores of different methods over different numbers of MCMC steps. We can see that with the same number of Langevin steps, the CoopFlow can generate much more realistic image patterns than the other two baselines. The results show that the CoopFlow can use less number of Langevin steps (i.e., a shorter Langevin flow) to achieve better performance.

| Model      # of MCMC steps | 10 | 20 | 30 | 40 | 50 | 200 |
|---|---|---|---|---|---|---|
| Short-run EBM | 421.3 | 194.88 | 117.02 | 140.79 | 198.09 | 54.23 |
| CoopNets | 33.74 | 33.48 | 34.12 | 33.85 | 42.99 | 38.88 |
| CoopFlow(Pre) | 16.46 | 15.20 | 15.80 | 16.80 | 15.64 | 17.94 |

Table 9: A comparison of FID scores of the short-run EBM, the CoopNets and the CoopFlow under different numbers of Langevin steps on the CIFAR-10 dataset.

## A.11 NOISE TERM IN THE LANGEVIN DYNAMICS

While for the experiments shown in the main text, we completely disable the noise term $\delta\epsilon$ of the Langevin equation presented in Eq. (3) by following Zhao et al. (2021) to achieve better results, here we try an alternative way where we gradually decay the effect of the noise term toward zero during the training process. The decay ratio for the noise term can be computed by the following:

$$\text{decay ratio} = \max((1.0 - \frac{\text{epoch}}{K})^{20}, 0.0) \tag{10}$$

where $K$ is a hyper-parameter controlling the decay speed of the noise term. Such a noise decay strategy enables the model to do more exploration in the sampling space at the beginning of the training and then gradually focus on the basins of the reachable local modes for better synthesis quality when the model is about to converge. Note that we only decay the noise term during the training stage and still remove the noise term during the testing stage, including image generation and FID calculation. We carry out experiments on the CIFAR-10 and the SVHN datasets using the *CoopFlow(Pre)* setting. The results are shown in Table 10.

| Dataset | $K$ | FID (no noise) | FID (decreasing noise) |
|---------|-----|----------------|------------------------|
| CIFAR-10 | 30 | 15.80 | 14.55 |
| SVHN | 15 | 15.32 | 15.74 |

Table 10: FID scores for the CoopFlow models trained with gradually reducing noise term in the Langevin dynamics.

## A.12 COMPARISON BETWEEN COOPFLOW AND SHORT-RUN EBM VIA INFORMATION GEOMETRY

Figure 14 illustrates the convergences of both the CoopFlow and the EBM with a short-run MCMC starting from an initial noise distribution $q_0$. We define the short-run EBM in a more generic form (Xie et al., 2016) as follows

$$p_\theta(x) = \frac{1}{Z(\theta)} \exp[f_\theta(x)]q_0(x), \quad (11)$$

which is an exponential tilting of a known reference distribution $q_0(x)$. In general, the reference distribution can be either the Gaussian distribution or the uniform distribution. When the reference distribution is the uniform distribution, $q_0$ can be removed in Eq. (11). Since the initial distribution of the CoopFlow is the Gaussian distribution $q_0$, which is actually the prior distribution of the normalizing flow. For a convenient and fair comparison, we will use the Gaussian distribution as the reference distribution of the EBM in Eq. (11). And the CoopFlow and the baseline short-run EBM will use the same EBM defined in Eq. (11) in their frameworks. We will use $\bar{p}_{\bar{\theta}}$ to denote the baseline short-run EBM and keep using $p_\theta$ to denote the EBM in the CoopFlow.

There are three families of distributions: $\Omega = \{p : \mathbb{E}_p[h(x)] = \mathbb{E}_{p_{\text{data}}}[h(x)]\}$, $\Theta = \{p_\theta(x) = \exp(\langle \theta, h(x) \rangle)q_0(x)/Z(\theta), \forall \theta\}$, and $A = \{q_\alpha, \forall \alpha\}$, which are shown by the red, blue and green curves respectively in Figure 14, which is an extension of Figure 1 by adding the following elements:

- $q_0$, which is the initial distribution for both the CoopFlow and the short-run EBM. It belongs to $\Theta$ because it corresponds to $\theta = 0$. $q_0$ is just a noise distribution thus it is far under the green curve. That is, it is very far from $q_{\alpha*}$ because $q_{\alpha*}$ has been already a good approximation of $p_{\theta*}$.

- $g_{\alpha*}$, which is the learned transformation of the normalizing flow $q_\alpha$, and is visualized as a mapping from $q_0$ to $q_{\alpha*}$ by a directed brown line segment.

- The MCMC trajectory of the baseline short-run EBM $\bar{p}_{\bar{\theta}}$, which is shown by the yellow line on the right hand side of the blue curve. The solid part of the yellow line, starting from $q_0$ to $\bar{\pi}^* = \mathcal{K}_{\bar{\theta}*}q_0$, shows the short-run non-mixing MCMC starting from the initial Gaussian distribution $q_0$ in $\Theta$ and arriving at $\bar{\pi}^*$ in $\Omega$. The dotted part of the yellow line is the potential long-run MCMC trajectory that is unrealized.

By comparing the MCMC trajectories of the CoopFlow and the short-run EBM in Figure 14, we can find that the CoopFlow has a much shorter MCMC trajectory than that of the short-run EBM, since the normalizing flow $g_{\alpha*}$ in the CoopFlow amortizes the sampling workload for the Langevin flow in the CoopFlow model.

## A.13 MORE CONVERGENCE ANALYSIS

The CoopFlow algorithm simply involves two MLE learning algorithms: (i) the MLE learning of the EBM $p_\theta$ and (ii) the MLE learning of the normalizing flow $q_\alpha$. The convergence of each of the two

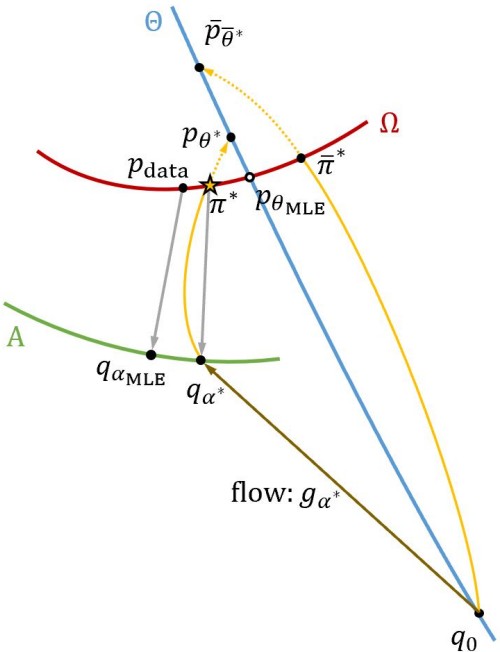

Figure 14: A comparison between the CoopFlow and the short-run EBM.

learning algorithms has been well studied and verified in the existing literature, e.g., Younes (1999); Xie et al. (2016); Kingma & Dhariwal (2018). That is, each of them has a fixed point. The only interaction between these two MLE algorithms in the proposed CoopFlow algorithm is that, in each learning iteration, they feed each other with their synthesized examples and use the cooperatively synthesized examples in their parameter update formulas. To be specific, the normalizing flow uses its synthesized examples to initialize the MCMC of the EBM, while the EBM feeds the normalizing flow with its synthesized examples as training examples. The synthesized examples from the Langevin flow are considered the cooperatively synthesized examples by the two models, and are used to compute their learning gradients. Unlike other amortized sampling methods (e.g., Han et al. (2019); Grathwohl et al. (2021)) that uses variational learning, the EBM and normalizing flow in our framework do not back-propagate each other through the cooperatively synthesized examples. They just feed each other with some input data for their own training algorithms. That is, each learning algorithm will still converge to a fixed point.

Now let us analyze the convergence of the whole CoopFlow algorithm that alternates two maximum likelihood learning algorithms. We first analyze the convergence of the objective function at each learning step, and then we conclude the convergence of the whole algorithm.

**The convergence of CD learning of EBM.** The learning objective of the EBM is to minimize the KL divergence between the EBM $p_\theta$ and the data distribution $p_{\text{data}}$. Since the MCMC of the EBM in our model is initialized by the normalizing flow $q_\alpha$, it follows a modified contrastive divergence algorithm. That is, at iteration $t$, it has the following objective,

$$\theta^{(t+1)} = \arg\min_\theta \mathbb{D}_{\text{KL}}(p_{\text{data}}||p_\theta) - \mathbb{D}_{\text{KL}}(\mathcal{K}_{\theta^{(t)}} q_{\alpha^{(t)}}||p_\theta). \tag{12}$$

No matter what kind of distribution is used to initialize the MCMC, it will has a fixed point when the learning gradient of $\theta$ equals to 0, i.e., $L'(\theta) = \frac{1}{n}\sum_{i=1}^n \nabla_\theta f_\theta(x_i) - \frac{1}{n}\sum_{i=1}^n \nabla_\theta f_\theta(\tilde{x}_i) = 0$. The initialization of the MCMC only affects the location of the fixed point of the learning algorithm. The convergence and the analysis of the fixed point of the contrastive divergence algorithm has been studied by Hinton (2002); Carreira-Perpiñán & Hinton (2005).

**The convergence of MLE learning of normalizing flow.** The objective of the normalizing flow is to learn to minimize the KL divergence between the normalizing flow and the Langevin flow (or the EBM) because, in each learning iteration, the normalizing flow uses the synthesized examples

generated from the Langevin dynamics as training data. At iteration $t$, it has the following objective

$$\alpha^{(t+1)} = \arg\min_{\alpha} \mathbb{D}_{\text{KL}}(\mathcal{K}_{\theta^{(t)}} q_{\alpha^{(t)}} || q_{\alpha}), \tag{13}$$

which is a convergent algorithm at each $t$. The convergence has been studied by Younes (1999).

**The convergence of CoopFlow.** The CoopFlow alternates the above two learning algorithms. The EBM learning seeks to reduce the KL divergence between the EBM and the data, i.e., $p_{\theta} \rightarrow p_{data}$; while the MLE learning of normalizing flow seeks to reduce the KL divergence between the normalizing flow and the EBM, i.e., $q_{\alpha} \rightarrow p_{\theta}$. Therefore, the normalizing flow will chase the EBM toward the data distribution gradually. Because the process $p_{\theta} \rightarrow p_{\text{data}}$ will stop at a fixed point, therefore $q_{\alpha} \rightarrow p_{\theta}$ will also stop at a fixed point obviously. Such a chasing game is a contraction algorithm, therefore the fixed point of the CoopFlow exists. Empirical evidence also support our claim. If we use $(\theta^*, \alpha^*)$ to denote the fixed point of the CoopFlow, according to the definition of a fixed point, $(\theta^*, \alpha^*)$ will satisfy

$$\theta^* = \arg\min_{\theta} \mathbb{D}_{\text{KL}}(p_{\text{data}} || p_{\theta}) - \mathbb{D}_{\text{KL}}(\mathcal{K}_{\theta^*} q_{\alpha^*} || p_{\theta}), \tag{14}$$

$$\alpha^* = \arg\min_{\alpha} \mathbb{D}_{\text{KL}}(\mathcal{K}_{\theta^*} q_{\alpha^*} || q_{\alpha}). \tag{15}$$

The convergence of the cooperative learning framework (CoopNets) that integrates the MLE algorithm of an EBM and the MLE algorithm of a generic generator has been verified and discussed in Xie et al. (2020a). The CoopFlow that uses a normalizing flow instead of a generic generator has the same convergence property as that of the original CoopNets. The major contribution in our paper is to start from the above fixed point equation to analyze where the fixed point will be in our learning algorithm, especially when the MCMC is non-mixing and non-convergent. This goes beyond all the prior works about cooperative learning.

