# OpenReview forum: "A Tale of Two Flows: Cooperative Learning of Langevin Flow and Normalizing Flow Toward Energy-Based Model"
_ICLR.cc/2022/Conference — ICLR 2022 Poster_

### Official Review · Reviewer_DUki · 2021-11-02

**Correctness:** 3
**Technical Novelty And Significance:** 3
**Empirical Novelty And Significance:** 3
**Recommendation:** 6
**Confidence:** 3

**Main Review:**

This paper is strongly motivated by the short-run MCMC energy-based model [1]. The key difference between this work and [1] is the use of a normalizing flow model for a nice initialization to shorten MCMC steps even more.

The idea is sound and well-motivated, however, my main concern on this work is that it seems both the theoretical explanation and empirical results do not support such a motivation very clearly.

* First, the theoretical explanation reveals that the short-run MCMC CoopFlow $\pi^*$ is a moment matching estimator of $p_{data}$. However, the current version of the explanation does not clearly show that $q_{\alpha}$ initialization can yield a shorter transition path (i.e., lower MCMC steps) compared to the standard Langevin flow started from $p_{0}$ (which is the main motivation of this work). Of course, it can be deduced from the fact that $q_{\alpha}$ tries to mimic $\pi$ thus automatically $D(\pi, q_{\alpha})$ can be lower than $D(\pi, p_{0})$ for a certain discrepancy D in most cases; but it will be nice if the authors can discuss such a case in a more systematic way, with adding some intuitive comparison of Figure 1 with Figure 5 of [1].

* In addition, the current empirical results do not validate the authors’ motivation sufficiently well, because a detailed cost-vs-performance trade-off study is not presented here. Table 1 (a) only reveals that the proposed CoopFlow gets worse FID scores than, for example, NCSN++ or EBM-Diffusion. I don't think every newly proposed model should beat the best model. However, since the main advantage of CoopFlow is a fast generation (as well as training) owing to the short-run MCMC with the normalizing flow-based amortized sampler, the author should discuss such a cost perspective in detail. For example, how many iteration MCMC/SDE steps are required for other models, and how about generating samples with the same (and small) number of steps for all models? Currently, I can only conclude the CoopFlow outperforms each of its components (it is also not absolutely certain because the reference flow model, Flow++, seems to be not evaluated solely, thus the author should ), which is also a nice contribution but not greatly fascinating.

***

[1] Erik Nijkamp, Mitch Hill, Song-Chun Zhu, and Ying Nian Wu. Learning non-convergent nonpersistent short-run MCMC toward energy-based model. arXiv preprint arXiv:1904.09770, 2019.


**Summary Of The Paper:**

The authors propose a cooperative learning framework of normalizing flow and energy-based Langevin MCMC models (CoopFlow). Specifically, the normalizing flow suggests a better initialization for the Langevin flow, then the latter generates synthetic samples via short-run MCMC. While the Langevin flow is trained with a standard MCMC likelihood toward the data distribution, the normalizing flow chases the Langevin one by maximizing the likelihood of the synthetic samples. The proposed CoopFlow seems to outperform each of its components.

**Summary Of The Review:**

Overall, while the paper is interesting, the lack of justification for the authors’ motivation prevents me from recommending a clear acceptance for this paper. My current evaluation of this paper is borderline but can be changed after reading the author’s rebuttal.

***
Post-rebuttal: I appreciate the authors’ thorough responses to my questions. After reading the responses, I have tended to accept this work, thus raised the review score accordingly. However, this rating is only marginally above the borderline, thus I would not champion the paper for acceptance.

After the authors' follow-up response: I feel more comfortable accepting this work. I am sorry to keep my score as 6, mainly due to there is no review score of 7 in the system. Instead, I increased the significance score of this paper accordingly.

---

> ### Author Response · Authors · 2021-11-23
> **Response to reviewer DUki  (Part 1 of 2)**
>
> We thank the reviewer for pointing out that our paper is sound and well-motivated. We also appreciate your valuable comments to improve our work. We have provided a point-by-point response below and followed your suggestions to revise our paper. Please re-consider your rating and support our work. If you still have any other questions, feel free to let us known.
>
> **Q1. The current version of the theoretical explanation does not clearly show that $q_{\alpha}$ initialization can yield a shorter transition path (i.e., lower MCMC steps) compared to the standard Langevin flow started from $p_0$ (which is the main motivation of this work). It will be nice if the authors can discuss such a case in a more systematic way, with adding some intuitive comparison of Figure 1 with Figure 5 of [a].**
>
>
> As you pointed out, the main motivation of our paper is to learn a normalizing flow to fast initialize the short-run Langevin flow in the context of energy-based cooperative learning framework, so that the short-run Langevin flow can be much shorter. We want to say it is very straightforward that the added normalizing flow will certainly help shorten the number of Langevin steps that we need in the CoopFlow algorithm, no matter whether in the short-run non-convergent MCMC scenario or in the convergent MCMC scenario.
>
>
> The convergence analysis studied via information geometry in section 4.2 have pointed out that the CoopFlow algorithm is a chasing game, in which the normalizing flow $q_{\alpha}$ chases the EBM $p_{\theta}$ toward $p_{\text{data}}$ because the learning gradient of EBM is to reduce the KL distance between EBM and data, while the MLE learning of normalizing flow will reduce the KL distance between normalizing flow and EBM. Since the  normalizing flow $q_{\alpha}$ keeps chasing the EBM (no matter whether the EBM is biased or not), the normalizing flow $q_{\alpha}$ will get closer and closer to the EBM $p_{\theta}$ during training.
>
> Therefore, at the convergent point $(\theta^{\star},\alpha^{\star})$, the Langevin flow starting from the normalizing flow, i.e.,$K_{\theta^{\star}}q_{\alpha^{\star}}$,  can be much closer to EBM $p_{\theta^{\star}}$ than a Langevin flow starting from the initial noise distribution, i.e., $K_{\theta^{\star}}q_0$, because the normalizing flow $q_{\alpha^{\star}}$ itself is already a reasonable approximation to EBM $p_{\theta^{\star}}$. In other words, it will take more Langevin steps if the Langevin starting from noise $q_0$ than that starting from a normalizing flow $q_{\alpha}$ that seeks to approximate the EBM. That is, we amortize the sampling workload to the normalizing flow. Therefore, the number of Langevin steps can be reduced without losing sampling capacity.
>
>
> Both Figure 1 in our paper and Figure 5 of [a] illustrates the convergence in the short-run MCMC case by information geometry. The major difference is that in our Figure 1 we have another parameter space $A$, which is for the normalizing flow $q_{\alpha}$. The short-run MCMC in our Figure 1 starts from $q_{\alpha{^\star}}$, while the short-run MCMC in Figure 5 of [a] starts from the initial distribution $q_0$. The point of the initial $q_0$ should be far under the green curve in our Figure because $q_0$ is just a noise distribution, which is very far from $q_{\alpha^{\star}}$ ($q_{\alpha^{\star}}$ has been already a good approximation of $p_{\theta^{\star}}$, while $q_0$ is just noise).  That means, the MCMC starting from $q_0$ need much more efforts (number of Langevin steps) to arrive at the destination than the MCMC starting from $q_{\alpha^\star}$.
>
> We try to deliver the above message by using an illustration, which has been included in our revision. **Please check the newly added subsection "A10 Comparison between CoopFlow and  short-run EBM via information geometry" in the appendix** for a formal explanation.
>
>
> === reference =====
>
> [a] Learning Non-Convergent Non-Persistent Short-Run MCMC Toward Energy-Based Model. NeurIPS 2019.

---

> ### Author Response · Authors · 2021-11-23
> **Response to reviewer DUki (Part 2 of 2)**
>
>
> **Q2. Since the main advantage of CoopFlow is a fast generation, the author should discuss such a cost perspective in detail. For example, how many iteration MCMC/SDE steps are required for other models, and how about generating samples with the same (and small) number of steps for all models?**
>
> Thank you for your valuable suggestion. We have followed your suggestions and compared our method with closely related baselines in Table 4.1 from the perspective of computational cost in terms of the number of MCMC steps for EBM.
>
> We compare our methods with two closely related baselines, (1) the single EBM with short-run MCMC starting from noise distribution [a], and (ii) Cooperative learning of an EBM and a generic generator [b]. The numbers of MCMC steps and FID reported in Table 4.1 are collected from the original papers.
>
>
> Table 4.1: A comparison of computational costs in terms of number of MCMC steps.
>
> |            Method            | number of MCMC steps | FID |
> |:----------------------------:|----------------------|-----|
> | CoopFlow = EBM + flow (ours) |           30         |15.80|
> | EBM with short-run MCMC  [a]    |           100        |48.21|
> | CoopNet = EBM + generator [b]   |           10         |33.61|
>
> In Table 4.2, we also report the FID performance of different related baseline methods under the different numbers of MCMC steps. We can see that with the same number of Langevin steps, the CoopFlow can generate much more realistic image patterns than the other two baselines. The quality of the generated images are evaluated by FID. The results of the experiments show that the proposed CoopFlow can use less number of MCMC steps to achieve better results.
>
> Table 4.2: A comparison of FID scores under different numbers of MCMC steps.
>
> |            Method            | 10    | 20     | 30     | 40     | 50     | 200   |
> |:----------------------------:| ----- | ------ | ------ | ------ | ------ | ----- |
> | CoopFlow = EBM + flow (ours) | 16.46 | 15.20  | 15.80  | 16.80  | 15.64  | 17.94 |
> |   EBM with short-run MCMC    | 421.3 | 194.88 | 117.02 | 140.79 | 198.09 | 54.23 |
> |  CoopNet = EBM + generator   | 33.74 | 33.48  | 34.12  | 33.85  | 42.99  | 38.88 |
>
>
> We have followed you suggestion to include the above additional experiments in our revised paper. **Please check a newly added subsection "A7. Analysis of the number of Langevin steps" in the Appendix.**
>
>
> ================= reference =======
>
> [a] Learning Non-Convergent Non-Persistent Short-Run MCMC Toward Energy-Based Model. NeurIPS 2019
>
> [b] Cooperative Training of Descriptor and Generator Networks. PAMI 2018.

---

> ### Author Response · Authors · 2021-11-28
> **Looking forward to your feedback**
>
> Dear Reviewer DUki,
>
> We have followed your suggestions to use both geometry information and empirical results to show that our method can reduce the number of Langevin steps. These modifications can better support the motivation of our paper.
>
> Moreover, we added quantitative results for the experiment of image reconstruction in section "A5 Quantitative Results for the Image Reconstruction Experiment". We added Figure 12 in Appendix to show evidence of the convergence of our method. We added a new subsection "A.8 More Convergence Analysis" in Appendix to discuss the convergence of the algorithm. We added a new subsection "A.9 Discussion on the bias of the Contrastive Divergence Objective for EBM" in Appendix to discuss the bias of the contrastive divergence and how this affect our model.
>
> Please kindly let us know if we have addressed your concerns, and feel free to tell us if you still have other questions. We hope you can re-consider your rating.
>
> Thanks a lot to help improve our paper.
>
> best,
>
> Authors

---

> > ### Comment · Reviewer_DUki · 2021-11-29
> > **Reply to the authors' response**
> >
> > I appreciate the authors’ thorough responses to my questions. After reading the responses, I have tended to accept this work, thus raised the review score accordingly. However, this rating is only marginally above the borderline, thus I would not champion the paper for acceptance.
> >
> > Regarding Q1, thank you for the detailed explanation and visualization for the proposed algorithm. It satisfies enough of my concern, although within the expectation.
> >
> > Regarding Q2, thank you for the valuable experiment, which supports the computational efficiency of the proposed CoopFlow compared to the CoopNet. While there is no direct comparison with NCSN++ or EBM-Diffusion (whose FID is better than CoopFlow), I am not dissatisfied with it, however, I’d like to follow-up on:
> > * Why the normalizing flow initialization is better than the generic generator used in CoopNet?
> > * Can the normalizing flow initialization and cooperative learning framework be used for improving SOTA models, e.g., score-based SDE (NCSN++ in Table 1 (a))?

---

> > > ### Author Response · Authors · 2021-11-29
> > > **Reply to the reviewer DUki's follow-up questions (Part 1 of 3)**
> > >
> > > Thank you for your reply and raising the review score after reading our reply. We would like to answer your follow-up questions below.
> > >
> > > **Q1: Thank you for the valuable experiment, which supports the computational efficiency of the proposed CoopFlow compared to the CoopNet. While there is no direct comparison with NCSN++ or EBM-Diffusion (whose FID is better that CoopFlow), I am not dissatisfied with it, however, I’d like to follow-up on:**
> > >
> > >
> > > In our previous reply, to better support the motivation of our method (i.e., the flow initialization can help reduce the number of MCMC steps), we make a fair comparison with an EBM with the same structure and a CoopNet model with the same EBM structure. That was why we don't include other types of generative models, e.g., score-based model and EBM-diffusion. **The score-based model (NCSN++) neither have a parameterized EBM nor have an MCMC that samples from a distribution**. It builds stochastic differential equation (SDE) to transform the data distribution to the noise, and the corresponding reverse SDE from noise to data is the resulting generator. NCSN++ belongs to another type of generative models, that is diffusion model, which is not directly related to our work. The EBM-diffusion baseline is also a diffusion model, which is a diffusion recovery likelihood method to tractably learn and sample from **a sequence of EBMs** trained on increasingly noisy versions of a dataset. Each EBM is trained with recovery likelihood, which maximizes the conditional probability of the data at a certain noise level given their noisy versions at a higher noise level. Both NCSN++ and EBM-diffusion models are heavy-duty models, and their compelling STOA performance need a lot of GPUs for training.
> > >
> > > To summarize, there is no need or unfair to compare our methods with NCSN++ and EBM-diffusion under different numbers of MCMC steps, since the NCSN++ doesn't have MCMC and the EBM-diffusion uses a sequence of EBMs for diffusion training. Even though the EBM-diffusion [a] use a sequence of EBMs, we can still report its performance with the total number of MCMC steps they use if you are interested in taking a look.
> > >
> > > The following is the updated Table 4.1 in our previous reply. We add the EBM-diffusion baselines with their MCMC steps (The last three lines in the Table 4.1). We use $T$ to denote the number of EBM used in the EBM-diffusion model, and $K$ to denote the number of MCMC steps for each EBM. Therefore, the total number of MCMC steps for the EBM-diffusion is $T \times K$. The results are directly corrected from the Table 2 in the original paper [a]. We can show that the the gain of EBM-diffusion is mainly obtained from $T>1$, that is, using more than one model for diffusion training. We believe our CoopFlow will also be improved by adding the diffusion training scheme by using a sequence of EBMs. All in all, the following results have shown that the proposed flow initialization strategy do improve the computational efficiency of training a single EBM.
> > >
> > > **We can add the following table and the above discussion in our revision. We hope you will feel satisfied on our current reply, and re-consider your rating again. Thanks.**
> > >
> > >
> > > Table 4.1: A comparison of computational costs in terms of number of MCMC steps.
> > >
> > > |            Method             | number of MCMC steps | FID   |
> > > |:-----------------------------:| -------------------- | ----- |
> > > | CoopFlow = EBM + flow (ours)  | 30                   | 15.80 |
> > > | EBM with short-run MCMC  [a]  | 100                  | 48.21 |
> > > | CoopNet = EBM + generator [b] | 10                   | 33.61 |
> > > |         EBM-diffusion         | 180   (T=1, K=180)   | 32.12 |
> > > |         EBM-diffusion         | 180   (T=6, K=30)    | 9.58  |
> > > |         EBM-diffusion         | 300   (T=6, K=50)    | 9.36  |
> > >
> > >
> > > [a] LEARNING ENERGY-BASED MODELS BY DIFFUSION RECOVERY LIKELIHOOD. ICLR 2021.

---

> > > ### Author Response · Authors · 2021-11-29
> > > **Reply to the reviewer DUki's follow-up questions (Part 2 of 3)**
> > >
> > >
> > > **Q2: Why the normalizing flow initialization is better than the generic generator used in CoopNet?**
> > >
> > > The advantage of using normalizing flow is two-folds:
> > >
> > > **(1) First of all,**
> > >
> > > The original CoopNet algorithm requires an MLE learning of the generator. If the generator is a generic generator, then its MLE requires time-consuming MCMC-based inference. That is, given a training set $\{\tilde{x}\}$, the learning gradient of a generic generator is given by
> > > \begin{align}
> > > \frac{\partial}{\partial \alpha} \log q_{\alpha}(\tilde{x})&=\frac{1}{q_{\alpha}(\tilde{x})} \frac{\partial}{\partial \alpha} \int q_{\alpha}(\tilde{x},z) dz\\\\
> > > & =\int \left[\frac{\partial}{\partial \alpha} \log q_{\alpha}(\tilde{x},z)  \right] \frac{q_{\alpha}(\tilde{x},z)}{q_{\alpha}(\tilde{x})}dz\\\\
> > > &= E_{q_{\alpha}(z|\tilde{x})}\left[ \frac{\partial}{\partial \alpha} \log q_{\alpha}(\tilde{x},z) \right]
> > > \end{align}
> > > where $q_{\alpha}(z|\tilde{x})$ is intractable posterior distribution, which can be evaluated by MCMC, such as Langevin dynamics that iterates
> > > \begin{align}
> > > z^{t+1} = z^{t} + \delta \frac{\partial}{\partial z} \log q_{\alpha}(z^{t}|\tilde{x}) + \sqrt{2\delta} \epsilon,  \epsilon \sim N(0,1)
> > > \end{align}
> > > where $t$ indexes the Langevin step. That is, for each training example $\tilde{x}$, we need MCMC-based inference to infer $z \sim p_{\alpha}(z|\tilde{x})$ so that we can compute the expectation term for the learning gradient of $\alpha$.
> > >
> > > One [a] might try to avoid the MCMC-inference part in training the generator by using VAE loss, which relies on an extra encoder as an inference network. That is, the framework need an EBM, a generic generator, and an encoder. Even though the inference become more efficient, but the number of parameters of the model increase. Moreover, the VAE loss is just a perturbation of MLE (i.e., lower bound), which is still biased to the true MLE.
> > >
> > > Therefore, the advantage of the choice of a normalizing low become obvious. We have a rigorous MLE for the flow training, we don't need MCMC inference, we don't require an extra encoder for variational learning.
> > >
> > > [a] Learning Energy-Based Model with Variational Auto-Encoder as Amortized Sampler. AAAI 2021.
> > >
> > > **(2) Secondly,**
> > > the above advantage is on the training side. In testing side and especially under the scenario of short-run non-mixing MCMC (actually this is the main focus in our paper), the normalizing flow can be much more convenient than the generic generator.
> > >
> > > In the experiment of reconstruction, we show that the short-run non-mixing MCMC can be considered as a flow-like latent variable generator, so that it, together with the normalizing flow, can be used for reconstruction. The following shows that the difference between the use of a generic generator and the use of a normalizing flow.
> > >
> > > Recall our model: $z \sim p_0(z); \hat{x}=g_{\alpha}(z); x=F_{\theta}(\hat{x},e)$, where $\theta$ is the Langevin flow, and $\alpha$ is normalizing flow, $p_0$ is Gaussian prior, and $e$ contains all the injected noises in the Langevin flow. (The reason why we can write the Langevin flow as a mapping $x=F_{\theta}(\hat{x},e)$ because it is a short-run non-convergent MCMC whose output $x$ is dependent on the initialization $\hat{x}$; otherwise $x$ will converge to the target distribution no matter where it starts. That is $x$ is independent on $\hat{x}$).
> > >
> > > The true latent space in the above model is $z$. In general, we should infer $z$ for image reconstruction, that is, given an image $x$, we find its latent code $z$ by using gradient descent on $L(z)=||x-F_{\theta}(g_{\alpha}(z),e)||^2$, with $z$ is initialized by $p_0$. However, $g$ is an invertible transformation. therefore we can infer $z$ by an efficient way, i.e., we can first find $\hat{x}$ by gradient descent on $L(\hat{x})=||x-F_{\theta}(\hat{x},e)||^2$, and then use $z=g_{\alpha}^{-1}(\hat{x})$ to get $z$. These two methods are equivalent, but the latter on is computationally efficient, since computing the gradient on the whole two-flow generator $F_{\theta}(g_{\alpha}(z),e)$ is time consuming.
> > >
> > > We need to point out that if $g$ is not the normalizing flow but a generic generator, we have to use the first method to infer $z$ for reconstruction. This is the advantage of using normalizing flow in our framework instead of other top-down generator. This might somehow answer your question about why choosing normalizing flow, because its invertibility will make the inference step more efficient than other alternatives.

---

> > > ### Author Response · Authors · 2021-11-29
> > > **Reply to the reviewer DUki's follow-up questions (Part 3 of 3)**
> > >
> > > **Q3: Can the normalizing flow initialization and cooperative learning framework be used for improving SOTA models, e.g., score-based SDE (NCSN++ in Table 1 (a))?**
> > >
> > > We have to say NCSN++ is a very different framework than EBM with MCMC and flow model. Comparing them is out of the scope in the current paper. However, your question is interesting, forward-looking, and even inspiring. The answer of your question is yes. Conceptually, normalizing flow initialization can be used to further reduce the complexity of the NCSN++ model. Additionally, the NCSN++ can be a stronger reversible transformation than normalizing flow for the cooperative learning framework, which might help to efficiently train more powerful EBMs. We will study these promising direction in our future works. Thanks.
> > >
> > > ---------------------------
> > > We hope our reply to your follow-up questions will further make you feel satisfied. We hope you can re-consider your score again and champion our work. Please feel free to keep asking more questions if you still have. Thanks you again for keeping improving our work.

---

> > > > ### Comment · Reviewer_DUki · 2021-11-30
> > > > **Thanks for the rapid and thorough reply**
> > > >
> > > > Thanks for the authors’ rapid and thorough reply.
> > > >
> > > > Especially, I deeply appreciate the authors have conducted an additional experiment with comparing their model with EBM-diffusion. I think this empirical evidence can support the efficiency of the proposed algorithm more clearly.
> > > >
> > > > Also, thank you for the justification of the normalizing flow initialization. I have agreed that the tractable and explicit likelihood estimation of flows would be helpful for convenient training of similar frameworks. As the authors pointed, MCMC likelihood might be biased when the number of MCMC steps is not sufficiently enough; on the other hand, ELBO might be a loose bound.
> > > >
> > > > Now, I feel more comfortable accepting this work. I am sorry to keep my score as 6, mainly due to there is no review score of 7 in the system. Instead, I increased the significance score of this paper accordingly.

---

> > > > > ### Author Response · Authors · 2021-11-30
> > > > > **Thank you for your virtual score of 7 (even though there is no such a score in the system)**
> > > > >
> > > > > Thank you for your positive feedback and timely response. Your professional review make us feel very impressive. Thanks again for your valuable suggestions and comments to improve our work. **You are an excellent and responsible ICLR reviewer!**

---

### Official Review · Reviewer_1uN8 · 2021-11-02

**Correctness:** 4
**Technical Novelty And Significance:** 3
**Empirical Novelty And Significance:** 3
**Recommendation:** 8
**Confidence:** 3

**Main Review:**

This paper provides a novel cooperative way to train the flow model and energy model. The idea is very attractive, and could be able to solve existing difficulties of generative models. My major concern for the CoopFlow comes from the fact that the training procedure cannot be interpreted as the minimization (or minimax) of a loss function. It is not clear whether the cooperative learning process is always convergent.

Some other comments are as follows:

1) The Langevin dynamics used in the paper is not an exact MCMC chain due to the time discretization. Is it possible to utitlize some MCMC samplers, e.g., Metropolis-adjusted Langevin algorithm in CoopFlow?

2) It is claimed that the KL div between \tilde p_\theta and p_\theta "decreases to zero monotonically". This conclusion is not widely known and requires reference.

3) Some important previous models and related references are omitted, including the deep energy model proposed by Liu within the stein variational framework, and the stochastic NF proposed by Noe consisiting of NF and MCMC.

4) Analysis in Section 4.2 is too confusing for the reader. This section is called "theoretical understanding" and there is no theorem. Can authors provide more understandable explanations?

**Summary Of The Paper:**

This paper develops a new generative model, called CoopFlow, by combining the energy-based model (EBM), normalizing flow (NF) and Langevin dynamics (LD). The CoopFlow updates all the parts in an cooperative manner: Parameters of NF are updated by using samples modified by LD, LD is performed based on the potential function provided by the EBM, and the gradient of log-likelihood of EBM are estimated according to samples generated by NF + LD. Here, LD can overcome the limitation of the model capacity of NF, and NF + LD can provide a more accurate estimation of the gradient of EBM loss. After learning, a powerful sampler and an accurate EBM can be obtained.

**Summary Of The Review:**

The article not only introduces some innovations, but also conducts a large number of experiments to demonstrate the superiority of the algorithm.

---

> ### Author Response · Authors · 2021-11-23
> **Response to Reviewer 1uN8 (Part 1 of 2)**
>
> We thank the reviewer for pointing out that the proposed method is novel and the conducted experiments are extensive. We also appreciate your  detailed review and valuable comments. We have provided a point-by-point response below and followed your suggestions to revise our paper. All the correction and newly added contents in our revised paper are in red color for your convenient check.
>
> **Q1. My major concern for the CoopFlow comes from the fact that the training procedure cannot be interpreted as the minimization (or minimax) of a loss function. It is not clear whether the cooperative learning process is always convergent.**
>
> The CoopFlow is a special case or a generalization of the cooperative learning framework via MCMC teaching [b]. The convergence of cooperative learning has been well studied and discussed in the original works of CoopNet [a][b] and also in its conditional generalization [c] and its variational generalization [d]. The experiments and empirical results also support the convergence of the algorithm. The results are extensive, including image synthesis [a,b,d], video synthesis[b], supervised conditional learning [c] and unsupervised image-to-image translation or video translation [e], etc. Therefore, it is very clear that the cooperative learning process can converge.
>
> Back to our CoopFlow algorithm that replaces the generic generator by a normalizing flow. The convergence of the CoopFlow is still well supported by the prior works. We need to point out that our work is different from the prior work not only because we use a special generator (i.e., normalizing flow) but also we study the short-run non-mixing MCMC case of EBM in the context of cooperative learning. This makes our current work connect to another theme, i.e., short-run MCMC as a generator [f].
>
> To be specific, our algorithm involves the MLE learning of EBM and MLE learning of normalizing flow. The convergence of each of the learning algorithms has been well studied. The only interaction between them is the **synthesized examples**, that is, the normalizing flow uses its synthesized examples to initialize the MCMC of the EBM, and the EBM will feed the flow with its synthesized examples as training data. That is, the EBM's objective is always to learn to converge to the data distribution (no matter how its MCMC is initialized), and the normalizing flow's objective is always to learn to converge to EBM (because it each learning iteration, the normalizing flow uses the EBM's synthesized examples as training data). The EBM MLE learning will reduce the KL distance between EBM and data, while the MLE learning of normalizing flow will reduce the KL distance between normalizing flow and EBM. Thus, the normalizing flow will chase the EBM toward the data. Since the data distribution is fixed, the chasing game is a contraction, then the fixed point exists. Empirical evidence has supported that our learning algorithm is a contraction.
>
> Our algorithm doesn't combine two models in a new designed loss and learn them by minimizing the loss. In that case, usually updating one model will need back-propagation going through the other one, then we need to carefully analyze if the new loss can converge before minimizing it. In our case, two well-known convergent algorithms just **cooperate** with each other. As long as two MLE algorithms themselves are convergent, their chasing game will converge to a fixed point.
>
> The above analysis is for the scenario where MCMC in the EBM MLE algorithm is mixing. What if the short-run MCMC case? The current literature about the short-run MCMC for EBM [f] has shown that even though with a non-mixing MCMC, the algorithm still converges to a moment matching estimator. That mean, if the short-run EBM can converge to a fix point, then the normalizing flow that chases the EBM will also converge to a fix point. That means the whole cooperative game will stop at a fix point. Our experiments also support this statement.
>
> We hope the above explanation can convince you that the cooperative learning algorithm is a contraction. That is, it has a fix point. Our paper mainly studies where this fix point will be by using information geometry.  Formally, we have revised our paper and added a new subsection "A8.More Convergence Analysis" in the appendix. Please read the formal version in our revised paper.
>
> [a] Cooperative Learning of Energy-Based Model and Latent Variable Model via MCMC Teaching. AAAI 2018.
>
> [b] Cooperative Training of Descriptor and Generator Networks. PAMI 2018.
>
> [c] Cooperative Training of Fast Thinking Initializer and Slow Thinking Solver for Conditional Learning. PAMI 2021
>
> [d] Learning Energy-Based Model with Variational Auto-Encoder as Amortized Sampler. AAAI 2021
>
> [e] Learning Cycle-Consistent Cooperative Networks via Alternating MCMC Teaching for Unsupervised Cross-Domain Translation. AAAI 2021
>
> [f] Learning Non-Convergent Non-Persistent Short-Run MCMC Toward Energy-Based Model, NeurIPS 2019.

---

> ### Author Response · Authors · 2021-11-23
> **Response to Reviewer 1uN8 (Part 2 of 2)**
>
> **Q2. The Langevin dynamics used in the paper is not an exact MCMC chain due to the time discretization. Is it possible to utitlize some MCMC samplers, e.g., Metropolis-adjusted Langevin algorithm in CoopFlow?**
>
> It is possible to use Metropolis-adjusted Langevin algorithm in CoopFlow. A complete implementation of Langevin Dynamics requires a momentum update and Metropolis-Hastings update in addition to $x_{t+1}=x_t + \frac{\delta^2}{2} \nabla_x f_{\theta}(x_t) + \delta \epsilon_t$, but most authors of recent EBM works find that these can be ignored in practice for small enough step size $\delta$. We use a simple implementation just for computational efficiency as other authors.
>
> **Q3. It is claimed that the KL div between $\tilde{p}_\theta$ and $p_\theta$ "decreases to zero monotonically". This conclusion is not widely known and requires reference.**
>
> Cover and Thomas's book "Elements of information theory" proved the law of thermodynamics. We have followed your suggestion and citted book in our revsion. Thanks.
>
> **Q4. Some important previous models and related references are omitted, including the deep energy model proposed by Liu within the stein variational framework, and the stochastic NF proposed by Noe consisiting of NF and MCMC.**
>
>
> We have followed your suggestion and cited the following references. They are very related. Thank you so much.
>
> [a] Stochastic Normalizing Flows. NeurIPS 2020
>
> [b] Qiang Liu et al. Learning Deep Energy Models: Contrastive Divergence vs. Amortized MLE. ArXiv 2017.
>
>
> **Q5. Analysis in Section 4.2 is too confusing for the reader. This section is called "theoretical understanding" and there is no theorem. Can authors provide more understandable explanations?**
>
> Thank you for your suggestion. We have changed the title of dection 4.2 in our revision by using "Understanding the Learned Two Flows" to avoid confusion.

---

> ### Author Response · Authors · 2021-11-28
> **Any follow-up questions?**
>
> Dear Reviewer 1uN8
>
> Thanks you again for your comments and supports.
>
> We have tried our best to address all your concerns. Besides, we have also substantially revised our paper by following other valuable suggestions from other reviewers. Please read the summary of our revision in
>
> https://openreview.net/forum?id=31d5RLCUuXC&noteId=jbEQbtFOzPi
>
> Please kindly let us know if you still have other follow-up questions, we will be more than happy to answer.
>
> best,
>
> Authors

---

> > ### Comment · Reviewer_1uN8 · 2021-11-30
> > **Response**
> >
> > After reading the reply by authors and comments of the other reviewers. The revision is fine for me, and I'll not change my score.

---

> > > ### Author Response · Authors · 2021-11-30
> > > **Thank you**
> > >
> > > Thank you for your reply !

---

### Official Review · Reviewer_bAHo · 2021-11-02

**Correctness:** 3
**Technical Novelty And Significance:** 3
**Empirical Novelty And Significance:** 3
**Recommendation:** 6
**Confidence:** 4

**Main Review:**


Strength
- The idea of the paper is well simple, and it seems to work well to generate realistic samples.
- It makes progress with respect to Cooperative learning via MCMC teaching, by using the normalizing flow instead of other kind of generators.

Weakness
- The theoretical validity of the generator is not very clear based on Section 4.2. In particular, there is no condition  or evidence to show that the generator converges to a fixed point in (8) and (9). Why there exists a fixed point?
- If possible, the baseline in Table 1 should also include the result of Ho2019 (used in the proposed model) in the category of Flow, to see if the MCMC teaching is important.

Questions:
- It is not clear what is evaluated in Section 5, is it the training set or test set? Can MSE score (often used in image reconstruction) be provided for Figure 3?
- What would happen if you use 200-step  instead of 30-step in CoopFlow (Pre) in section 5.2?

Typo
- section 4.1, in practise -> in practice

**Summary Of The Paper:**

This paper proposes to train an energy-based model with a normalizing flow to achieve rapid and high-quality MCMC sampling.
It shows the trained CoopFlow is capable of synthesizing toy data and realistic images (reasonable compared to many baseline methods in terms of common evaluation metrics).  Applications to image reconstruction and interpolation are also included.

**Summary Of The Review:**

This paper introduced a simple idea to train an energy-based model, which is very challenging in high-dimensions. The numerical result makes quite significant progress over state-of-the-art methods in terms of FID scores on natural images.

---

> ### Author Response · Authors · 2021-11-23
> **Response to Reviewer bAHo (Part 1 of 2)**
>
> We thank the reviewer for valuable comments. We have provided a point-by-point response below and followed your suggestions to revise our paper. Please re-consider your rating and support our work. Our revised paper have been improved substantially by including all the useful suggestions from the reviewers and extra experimental results. The corrected parts and the newly added contents in the revised paper are marked in red color so that you can check them conveniently.
>
> **Q1. There is no condition or evidence to show that the generator converges to a fixed point in (8) and (9). Why there exists a fixed point?**
>
> The CoopFlow algorithm simply involves two MLE learning algorithms: (i) the MLE learning of the EBM $p_{\theta}$ and (ii) the MLE learning of the normalizing flow $q_{\alpha}$. The convergence of each of the two learning algorithms has been well studied and verified in the existing literature, e.g., [a][b][c]. That is, each of them has a fixed point. The only interaction between these two algorithms in the proposed CoopFlow algorithm is that, in each iteration, they feed each other with their synthesized examples and use the cooperatively synthesized examples in their parameter update formulas. To be specific, the normalizing flow uses its synthesized examples to initialize the MCMC of the EBM, and the EBM feed the normalizing flow with its synthesized examples as training examples. The synthesized examples from the Langevin flow are considered the cooperatively synthesized examples by two models, and used to compute their learning gradients. Unlike other amortized sampling methods (e.g., [d][e]) that uses variational learning, the EBM and normalizing flow in our framework don't back-propagate each other through the cooperatively synthesized examples. They just feed each other with some input data for their own training algorithms. That is, each learning algorithm will still converge and have a fixed point.
>
> Now let us analyze the convergence of the whole CoopFlow algorithm that alternates two MLE algorithms. We first analyze the convergence of the objective function at each learning step, and then we conclude the convergence of the whole algorithm.
>
> **The convergence of CD learning of EBM**. The learning objective of the EBM is to minimize the KL distance between the EBM and the data distribution. Since the MCMC of EBM in our case is initialized by the normalizing flow, it follows a modified contrastive divergence algorithm. That is, at iteration $t$, it has the following objective,
> \begin{align}
> \theta^{(t+1)} = \arg \min_{\theta}
> D_{\text{KL}}(p_{\text{data}}||p_{\theta})-D_{\text{KL}}(K_{\theta^{(t)}} q_{\alpha^{(t)}}|| p_{\theta}).
> \end{align}
> No matter what kind of distribution is used to initialize the MCMC, it will has a fixed point when the learning gradient of $\theta$ equals to 0, i.e., $L'(\theta)= \frac{1}{n}\sum_{i=1}^{n}\nabla_{\theta}f_{\theta}(x_i)-\frac{1}{n}\sum_{i=1}^{n}\nabla_{\theta}f_{\theta}(\tilde{x}_i)=0$. The initialization of the MCMC only affects the location of the fixed point of the learning algorithm. The convergence and the analysis of the fixed point of contrastive divergence has been studied by papers [f][g].
>
> **The convergence of MLE learning of normalizing flow** The normalizing flow’s objective is to learn to minimize the KL divergence between the normalizing flow and the Langevin flow because, in each learning iteration, the normalizing flow uses the EBM’s synthesized examples as training data. At iteration $t$, it has the following objective
> \begin{align}
> \alpha^{(t+1)} = \arg \min_{\alpha} D_{\text{KL}}(K_{\theta^{(t)}} q_{\alpha^{(t)}}|| q_{\alpha} ),
> \end{align}
> which is a convergent algorithm at each $t$. The convergence has been studied by [a].
>
> **(The reply of this question Q1 is unfinished and it is to be continued in the "Second part of reply for Q1")**

---

> > ### Author Response · Authors · 2021-11-23
> > **Second part of reply for Q1**
> >
> > **(this is the second part of reply for Q1)**
> >
> > **The convergence of CoopFlow** The CoopFlow alternates the above two learning algorithms. The EBM learning will reduce the KL distance between EBM and data, i.e., $p_{\theta} \rightarrow p_{data}$; while the MLE learning of normalizing flow will reduce the KL distance between normalizing flow and EBM, i.e., $q_{\alpha} \rightarrow p_{\theta}$. Therefore, the normalizing flow will chase the EBM toward the data distribution gradually. Because the process $p_{\theta} \rightarrow p_{\text{data}}$ will stop at a fixed point, therefore $q_{\alpha} \rightarrow p_{\theta}$ will also stop at a fixed point obviously. Such a chasing game is a contraction algorithm, therefore the fixed point of CoopFlow exists. Empirical evidence has also supported that our EM-like learning algorithm is a contraction. If we use $(\theta^*,\alpha^*)$ to denote the fixed point of the CoopFlow, according to the definition of a fixed point, $(\theta^*,\alpha^*)$  will satisfy
> > \begin{align}
> > \theta^{\star}&= \arg \min_{\theta}D_{\text{KL}}(p_{\text{data}}||p_{\theta})-D_{\text{KL}}(K_{\theta^{\star}} q_{\alpha^{\star}}|| p_{\theta} ) \\\\
> > \alpha^{\star} &= \arg \min_{\alpha} D_{\text{KL}}(K_{\theta^{\star}} q_{\alpha^{\star}}|| q_{\alpha}).
> > \end{align}
> > The convergence of the cooperative learning framework (CoopNets) that integrates MLE algorithm of an EBM and MLE algorithm of a generic generator has been verified and discussed in the original paper of CoopNets [h]. The CoopFlow that uses a normalizing flow instead of a generic generator has the same convergence property as that of the original CoopNets. The major contribution in our paper is to start from the above fixed point equation to discuss where the fixed point will be in our algorithm when the MCMC of the EBM is non-mixing and non-convergent. All the existing cooperative learning papers do not study the scenario of short-run MCMC. Our paper goes beyond all the prior works about cooperative learning.
> >
> > **We have followed you suggestion to add the above convergence analysis in the newly added subsection "A8. More Convergence Analysis" in the appendix of our revision. Please take a look!**
> >
> > ================== reference ===================
> >
> > [a] Younes, L. 1999. On the convergence of markovian stochastic algorithms with rapidly decreasing ergodicity rates. Stochastics: An International Journal of Probability and Stochastic Processes 65(3-4):177–228.
> >
> > [b] Diederik P. Kingma. Glow: Generative Flow with Invertible 1x1 Convolutions.
> >
> > [c] A Theory of Generative ConvNet. ICML 2016
> >
> > [d] Divergence Triangle for Joint Training of Generator Model, Energy-based Model, and Inference Model. CVPR 2018
> >
> > [e] No MCMC for me: Amortized sampling for fast and stable training of energy-based models. ICLR 2021.
> >
> > [f] Geoffrey E. Hinton. Training products of experts by minimizing contrastive divergence. Neural Computation, 2002
> >
> > [g] Miguel and  Geoffrey  E.  Hinton. On  contrastive  divergence  learning.    AISTATS, 2005.
> >
> > [h] Xie et al. Cooperative Training of Descriptor and Generator Networks. TPAMI 2018.

---

> > > ### Comment · Reviewer_bAHo · 2021-11-24
> > > **Q1**
> > >
> > > It is not very clear to me why " such a chasing game is a contraction algorithm, therefore the fixed point of CoopFlow exists. ", do you have a more clear statement?  Where in the paper [h] that the convergence of the cooperative learning framework (CoopNets) has been verified?

---

> > > > ### Author Response · Authors · 2021-11-25
> > > > **Reply to follow-up questions from Reviewer bAHo**
> > > >
> > > > **Q1 follow-up 1: It is not very clear to me why " such a chasing game is a contraction algorithm, therefore the fixed point of CoopFlow exists. ", do you have a more clear statement?**
> > > >
> > > > Thank you for your follow-up question. We would like to explain to you in another way. We have three distributions $p_{data}$, $p_{ebm}$, $p_{flow}$, where $p_{data}$ is the fixed target destination, and $p_{ebm}$ are $p_{flow}$ are movable. At the beginning, they are far from $p_{data}$.
> > > >
> > > > In this chasing game, $p_{ebm}$ goes to its destination $p_{ebm}$, this is a contraction, because this is a minimization of KL distance.  $p_{ebm^{\star}}$ exists. $p_{flow}$'s behavior will not affect the property of contraction of $p_{ebm}$, but it will affect the location of the convergent point of $p_{ebm^{\star}}$. So we certainly have $p_{ebm^{\star}}$ that stops at the point whose learning gradient equals to zero $L'=0$, that is.
> > > > \begin{align}
> > > > \frac{1}{n}\sum_{i=1}^{n}\nabla_{\theta^*}f_{\theta^*}(x_i)=\frac{1}{n}\sum_{i=1}^{n}\nabla_{\theta^*}f_{\theta^*}(\tilde{x}_i)
> > > > \end{align}
> > > > (No matter what the synthesis $\tilde{x}$ are, there must be a fixed point $\theta^*$ to make the gradient equals to zero. This is contraction.)
> > > >
> > > > As to the follower $p_{flow}$, whose destination is $p_{ebm}$. That is, $p_{flow}$ tries to minimize the distance between $p_{flow}$ and $p_{ebm}$. Since $p_{ebm}$ will eventually converges to a fixed point, $p_{flow}$ will eventually converge to a fixed point too. With a fixed $p_{EBM}$, learning flow $p_{flow}$ by minimizing the distance between the fixed EBM and $p_{flow} is a contraction.
> > > >
> > > > An ideal case of the chase game is, $p_{ebm}$ arrives at its destination $p_{data}$ (i.e., fixed point), then $p_{flow}$ catches up with $p_{ebm}$ and arrived at $p_{ebm}$'s position (fixed point). Both $p_{ebm}$ and $p_{flow}$ will not change any more.
> > > >
> > > >
> > > > A special implementation (discretized version) of the algorithm is that, we first let EBM go first with enough iterations until it converges, EBM will stops at a fixed point when gradient equals to zero (MLE or CD's convergence property). Then we let the flow go with enough iterations until it converges, flow will catch up with the EBM's distribution and stop (flow's MLE convergence property). The whole algorithm is a contraction rather than divergency. In CoopFlow, alternating the two MLE algorithms, without letting each of them run too many iterations at a time, can allow the $p_{flow}$ better follow the $p_{EBM}$ in practice, since each time we only allow short-run MCMC.
> > > >
> > > > Different from min-max, where the objective looks like contradiction, our algorithm have both algorithms that minimizes the KL. The contraction is obvious. The newly added learning curves in Figure 12 in Appendix are also empirical evidences of the contraction and convergence of the method. The FID score we get in image generation (i.e., density estimation) is also a contraction evidence.
> > > >
> > > > One more intuitive explanation. person A starting from Beijing and person B starting from Tokyo. person A's destination is New York, and person A is looking at the map (i.e., training data) and heading for New York. Person B's goal is finding A, and B is chasing A with a GPS showing B's position.  This process is a contraction and will converge. At the end both A and B will meet at New York.
> > > >
> > > >
> > > > **Q1: follow-up 2: Where in the paper [h] that the convergence of the cooperative learning framework (CoopNets) has been verified?**
> > > >
> > > > As to the reference [h], we just wanted to say: (1) all the empirical results on different tasks, including image, video synthesis, shown in [h] are evidence to verify the contraction. If it is not a contraction, (for example, maximizing a distance is not a contraction), no meaningful results can be obtained, that is, FID that measures distance between two distributions will not decrease to a small value. (2) [h] mentioned that the MLE algorithms of both EBM and generator are contraction, and pointed out that "The convergence of an algorithm of this type to the maximum likelihood estimate has been studied by [Younes 1999]". The chasing game built on such two contractions is obviously contraction. The EBM's behavior is a contraction, the flow's contraction depends on the EBM's contraction, thus the flow is contraction. [h] also studies where the convergent point will be in section "3.2 Theoretical understanding" to show that the convergent point will be at MLE solution. Our paper's contribution lies in the analysis of the convergent point with short-run non-mixing MCMC, which will be the moment matching estimator.
> > > >
> > > > Please let me know if you still have any other questions? We are happy to answer.  Happy Thanksgiving!
> > > >
> > > > ==============reference ================
> > > >
> > > > L. Younes, “On the convergence of markovian stochastic algorithms
> > > > with rapidly decreasing ergodicity rates,” Stochastics: An International
> > > > Journal of Probability and Stochastic Processes, vol. 65, no. 3-4, pp.
> > > > 177–228, 1999

---

> > > > ### Author Response · Authors · 2021-11-29
> > > > **emphasize again the convergence concern you might have and the major contribution of our paper**
> > > >
> > > > We would like to emphasize again the convergence concern you might have here
> > > >
> > > > The proposed CoopFlow is built on an existing cooperative learning framework [a,b,c] that are used for joint training an EBM and a generator. The normalizing flow in our framework is just a special generator. Analyzing the convergence of an existing learning scheme is out of scope of our current research. And we DO show that in our revision, the cooperative learning is based on the MLE algorithm of EBM and MLE algorithm of flow, both of them are well-known convergent algorithms. There is no need to prove the convergence of both MLE algorithms again. We believe the current discussion about the convergence in our revised paper is sufficient. The main contribution of our paper is to show that the cooperative learning with EBM using short-run non-mixing MCMC and a normalizing flow will lead to a stochastic normalizing flow (Langevin flow + normalizing flow) model whose estimation is actually a moment matching estimator.
> > > >
> > > > In our paper, we directly start from the "convergence equations" (Eq.8 and 9 in our paper) which are directly from the original paper [a]. We study, under the scenario of non-mixing MCMC, where the convergent point will be. It is unreasonable to degrade the paper because it doesn't prove the convergence of a prior work again that has been published in several premier journal papers or even a text book before. We hope you can focus more on our contribution on the short-run non-mixing MCMC, which is an underexplored research direction in the current EBM research.
> > > >
> > > > We hope you can re-consider your rating after reading again our revision, in which we have made substantial improvement. A summary of the improvement is listed in https://openreview.net/forum?id=31d5RLCUuXC&noteId=jbEQbtFOzPi
> > > >
> > > > ============reference===============
> > > >
> > > > [a] Xie et al. Cooperative Training of Descriptor and Generator Networks. (IEEE Transactions on Pattern Analysis and Machine Intelligence) 2018. https://arxiv.org/pdf/1609.09408.pdf
> > > >
> > > > [b] Adrian Barbu, Song-Chun Zhu. Monte Carlo Methods. Springer 2020. https://bit.ly/mc_book_sample
> > > >
> > > > [c] Xie et al. Cooperative Training of Fast Thinking Initializer and Slow Thinking Solver for Multi-Modal Conditional Learning. (IEEE Transactions on Pattern Analysis and Machine Intelligence) https://arxiv.org/pdf/1902.02812.pdf

---

> > > > ### Author Response · Authors · 2021-12-02
> > > > **Looking forward to your feedback**
> > > >
> > > > Dear bAHo,
> > > >
> > > > Please kindly let us know if you still need further clarification. We will be more than happy to answer.
> > > >
> > > > The major contribution of our paper is to study the interaction between a short-run stochastic Langevin flow and a deterministic normalizing flow in the cooperative learning framework (existing prior learning framework).
> > > >
> > > > Our paper found that the resulting generative model is a two-flow generator that converges to a moment matching estimator.  That is
> > > >
> > > > $E_{p_{data}} [ \nabla {\theta} f_{\theta^*}(x)] = E_{\pi^*(x)} [ \nabla {\theta} f_{\theta^*}(x)]$,
> > > >
> > > > where $\pi^*(x)$ is the distribution of our two-flow latent variable generator, i.e.,
> > > > $z \sim p0(z); \tilde{x} = g_{\alpha}(z); x = F_{\theta}(\tilde{x})$,
> > > > where $g$ is normalizing flow, $F$ is the short-run Langevin flow.
> > > >
> > > > Please focus more on the contribution and the significance of our paper that is working on the new concept of "non-convergent MCMC is a flow-like generator”, and champion our paper.
> > > >
> > > > Thank,
> > > >
> > > > Authors

---

> ### Author Response · Authors · 2021-11-23
> **Response to Reviewer bAHo (Part 2 of 2)**
>
> **Q2. If possible, the baseline in Table 1 should also include the result of Ho2019 (used in the proposed model) in the category of Flow, to see if the MCMC teaching is important.**
>
> The original paper of Ho2019 doesn't report their FID score. And we have done the ablation study of using our own implementation of the normalizing flow structure of Ho2019 (this implemetation has also been used as a part of our CoopFlow). The results are shown in Table 6 in the appendix. You can see the FID score we got is 92.10 for the normalizing flow only case, while the score for CoopFlow (even without pretraining the normalizing flow model) can reach 21.16. We don't include this results in Table 1 because we are not sure whether using the offcial implementation of Tensorflow might achieve a better result (we can not find a pretrained checkpoint in their website). But given that we have use the same implementation (and model strucuture) in both the normalizing flow only case, i.e. flow++(Ho2019) as well as the CoopFlow case, we believe it is fair to draw the conclusion that the Langevin Flow (i.e., MCMC teaching) is impoatant because it can improve the performance of the normalizing fow. Also, you can easily find the gap between flow++(Ho2019) and our CoopFlow by simply looking at the synthesized images on cifar10 provided by these two models. Comparing our Figure 4(a) and Ho2019's Figure 2(b) (or Ho2019's Figure 6 in the supplement), we can easily find that our generated samples are much better.
>
> **Q3. It is not clear what is evaluated in Section 5, is it the training set or test set?**
>
> We calculate the FID score using the samples we generate from the model with the training set data. While in the image reconstruction task, we carry our experments on the test set of CIFAR-10 dataset. For the inpainting and interpolation experiments, we use the CelebA dataset. Given this dataset usually doesn't have a training and testing slipt. We train the model on all the data and thus the inpainting and interpolation are done on training set. We have also modified our paper to make this more clear.
>
>
> **Q4. Can MSE score be provided for Figure 3?**
>
> We have followed your valuable suggestion to provide quantitaive evaluation in terms of MSE for the results shown in Figure 3. Following [a], we calculate the per pixel MSE on 1000 examples in the leave-out testset of CIFAR-10. We use a 200-step gradient descent to minimize the reconstruction loss. We report the final per pixel MSE in Table 2.1. For a baseline, we test the reconstruction of the same 1,000 images using the short-run EBM model which has the same EBM structure as that in our CoopFlow and uses 100 steps of MCMC. We can see the CoopFlow works better than the short-run EBM in this image reconstruction task.  Again, the purpose of the reconstruction experiments is to verify that our CoopFlow with a short-run non-mixing MCMC and a normazling flow is a latent variable generator that can be used for reconstruction. If the MCMC is mixing, then it can not be used for this purpose.
>
> Table 2.1: Reconstruction error (MSE per pixel).
>
> |     Method      | MSE    |
> |:---------------:| ------ |
> |  Short-run EBM  | 0.1083 |
> | CoopFlow (ours) | 0.0254 |
>
>
> [a] Learning Non-Convergent Non-Persistent Short-Run MCMC Toward Energy-Based Model. NeurIPS 2019.
>
>
> **Q5. What would happen if you use 200-step instead of 30-step in CoopFlow (Pre) in section 5.2?**
>
> We have followed your suggestion to use a 200-step Langevin dynamics in our CoopFlow (Pre) on Cifar10 dataset. We use the step size as the CoopFlow(Long). Due to limited time for carrying our experiments, currently, we get the fid as 17.94 in this setting. We believe after more tuning, this score may be better. On the other hand, this may also suggest that in the pretrained normalizing flow case, the initial proposal provied by normalizing flow may be good enough, so long chain MCMC may actually not be needed. We have included this result in Table 10 of our appendix. Table 10 also shows FID performacnes over different numbers of MCMC steps.
>
> **Q6. Typo: section 4.1, in practise -> in practice**
>
> We have fixed the typo. Thanks.

---

### Official Review · Reviewer_QoKP · 2021-11-02

**Correctness:** 3
**Technical Novelty And Significance:** 2
**Empirical Novelty And Significance:** 2
**Recommendation:** 3
**Confidence:** 4

**Main Review:**

The paper describes a reasonable approach to generative modeling and demonstrates promising metric values. However, the main issue of the paper is that the particular choice of the generative models is not well-motivated. That is, the first question that appears in the reader's mind is why do we need to combine a flow with a short-run chain? Couldn't we use any other pair of $C_n^2$ combinations (where $n$ is the number of different approaches) to learn an efficient generative model? Finally, couldn't we just increase the size of our baseline model instead of mixing it with another approach? Unfortunately, the paper doesn't address any of these questions. To be more precise, the authors do not provide any motivation for the usage of these particular models. The main argument in favor of the proposed approach is the empirical comparison, which is, however, not extensive, since the models are not compared in terms of computational budget.

Other comments:
- The proposed objective in equation (8) closely resembles the Contrastive Divergence objective as noted by the authors. The contrastive divergence objective is notorious for its biasedness since the samples from the model ($K_\theta q_\alpha$ here) are usually treated to be independent of parameters $\theta$. I wonder if the authors follow the same way for their objective. Unfortunately, the authors neither clarify this in the paper nor provide an analysis of the bias (in the case of the biased scheme).
- The ability to model the density is not demonstrated empirically. The authors mention that their approach is able to model the density of the learned distribution (which could be a very nice motivation by the way), but they do not validate this property empirically even on a toy example.
- There is no convergence analysis. The authors neither prove nor demonstrate empirically the convergence of the procedure. Moreover, according to the last paragraph of Section 4.2, the scheme performs "an infinite looping" hence does not converge. This non-convergence property is usually considered as a significant flaw of the approach (e.g. in GANs literature).
- Some empirical results are not trustworthy. Providing comparison with other generative models the authors report numbers from different papers. However, the comparison on Celeba is hardened by the fact that many papers do not downsample the images to 32x32 resolution as the authors do. For instance, NT-EBM (Nijkamp, 2020) uses 64x64 resolution and couldn't be compared in terms of FID to the proposed model.
- Reconstruction use case is not clear. The experiments with image reconstruction require additional clarification. It is not clear what kind of a problem the authors approach there. Are these images corrupted by some noise? The choice of the intermediate representation (the output space of the flow) for the latent space is also not motivated. Why couldn't one take the initial distribution that is an input for the flow?
- Overall, the numerous applications listed after the generative modeling experiments should be measured quantitatively rather than qualitatively by several examples.

**Summary Of The Paper:**

The paper proposes a generative model that consists of two components: a normalizing flow and an energy-based model with short-run MCMC as proposed in (Nijkamp, 2019). The whole procedure operates by pushing some simple distribution through the NF and then running the short-run chain on these samples. Both models are learned jointly as follows. The normalizing flow tries to capture the limiting distribution of the EBM by minimizing
$$\text{KL}(K_\theta q_\alpha \Vert q_\alpha),$$
where $q_\alpha$ is the density of the NF with parameters $\alpha$ and $K_\theta$ is the transition kernel of the short-run chain with parameters $\theta$. Simultaneously, the EBM is learned to capture the data by minimizing
$$\text{KL}(p_{\text{data}}\Vert p_\theta) - \text{KL}(K_\theta q_\alpha\Vert p_\theta).$$

In Section 4.2, the authors provide some analysis of the proposed procedure. Namely, the authors consider the perfect scenario (where the NF matches the EBM density and they both match data) and give some intuition for the case when the models are not expressive enough to perfectly match the data.

Finally, the authors provide an empirical study of the proposed procedure. They report metrics for image generation on CIFAR10, SVHN, Celeba (downsampled to 32x32); give several examples of image inpainting and latent space interpolation.

**Summary Of The Review:**

The paper describes a reasonable approach to generative modeling and demonstrates promising metric values. However, the approach itself is a combination of two well-known models. The main issue of the paper is that the particular choice of the generative models is not well-motivated.

---

> ### Author Response · Authors · 2021-11-23
> **Response to Reviewer QoKP (Part 1 of 8)**
>
> We thank the reviewer for the very detailed review and valuable comments. We have provided a point-by-point response below and have followed your suggestions to revise our paper. Our paper has been substantially improved by including all the suggestions from all reviewers as well as some new experiment results. The corrected and newly added contents in our revised paper are marked in red color, so that you can check them conveniently.
>
> **Q1. Motivation of combining Langevin flow and Normalizing flow**
>
> Our major motivation has been discussed in the last paragraph in our original submission. To be specific, our paper is mainly motivated by the current short-run MCMC EBM paper [a], which realizes that the convergent MCMC can not be guaranteed in learning EBM via MLE in practice and the synthesized examples are actually generated by a flow-like generator that is constructed by a short sequence of Langevin steps, i.e., a short-run MCMC. Although the short-run MCMC generator is appealing  due to its simplicity of implementation, its training is time-consuming because the short-run MCMC still needs to be built with roughly 100 Langevin steps to ensure a sufficient model capacity. **To further reduce the length of the short-run MCMC, we try to use a deterministic transformation, i.e., normalizing flow, to fast initialize the short-run MCMC so that the MCMC can be much shorter.**
>
> The above main motivation has been also pointed out by Reviewer DUki: "This paper is strongly motivated by the short-run MCMC energy-based model, and tries to use a normalizing flow model for a nice initialization to shorten MCMC steps even more."
>
> The motivation of choosing normalizing flow instead of a generic generator, is because the cooperative learning framework needs to incorporate an MLE algorithm of EBM and an MLE of generator. **The MLE of a normalizing flow generator will be much more tractable than the MLE of any other generic generator**. The latter might resort to either MCMC-based inference to evaluate the posterior distribution or another encoder network for variational inference.
>
> The normalizing flow model will also benefit from its cooperation with EBM  because its expressivity will be improved by the stochastic sampling process guided by the EBM. [b] has shown that stochasticity overcomes expressivity limitation of normalizing flows resulting from the invertibility constraint. Such a benefit also motivate us to propose the CoopFlow framework. Our experiments have shown that the CoopFlow indeed improves the performance of a single normalizing flow. Here we also want to grant credits to Reviewer 1uN8 who pointed out the related reference [b]. We have added it into our reference. [b] justifies advantage of the combination of stochastic Langevin flow and deterministic normalizing flow.
>
> **We have revised our paper by explicitly mentioning our motivation we discuss above. Please check the last paragraph in our revised paper. We use red color to highlight our revision.** Thank you for your constructive suggestion to improve the presentation of our paper.
>
>
> ========= reference===============
>
> [a] Learning Non-Convergent Non-Persistent Short-Run MCMC Toward Energy-Based Model. NeurIPS 2019.
>
> [b] Stochastic Normalizing flow. NeurIPS 2020.
>
> **Q2. Couldn't we just increase the size of our baseline model instead of mixing it with another approach?**
>
> Every model has its own advantage and shortcoming. Increasing the size of the model will not overcome their shortcoming. Increasing the size might also lead to (1) the difficulty of optimization due to highly non-convex parameter space and (2) the computation disaster due to a super big model. Combining two different types of models can overcome their own shortcomings and obtain the best of both worlds. In the proposed CoopFlow, the stochasticity from Langevin flow overcomes expressivity limitations of normalizing flows resulting from the invertibility constraint, whereas the normalizing flow that initializes the Langevin flow can improve efficiency of pure MCMC/Langevin flow. Our experiments have verified the above statement.

---

> ### Author Response · Authors · 2021-11-23
> **Response to Reviewer QoKP (Part 2 of 8)**
>
> **Q3. The empirical comparison is not extensive since the models are not compared in terms of computational budget.**
>
> In Table 1.1, we compare the computational budget in terms of model size (number of parameters) for our model and other closely related models, including flow-based models and those frameworks jointly train an EBM and a generator. We report models used on the Cifar10 dataset. We can see that our model achieves better FID with slightly more parameters than the EBM-FCE, which is a generative model that trains an EBM and a flow by noise constrative estimation. The "Flow++ only" is the normalizng flow used in our CoopFlow. "Flow++ only" uses exactly the same implementation as that in our CoopFlow. Thus, it shows the benefits (FID) by introducing an EBM to the normalzing flow (Flow++). On the other hand, we can see VAEBM uses a much larger pretrained generator model (NVAE) [a] than CoopFlow and its FID is better than ours. We have put these results in the newly added subsection **"A6. Comparison of model sizes" in the Appendix.**
>
>  Table 1.1: Comparison of FID and model size of different models.
>
> |     Method     | number of parameters | FID   |
> |:--------------:| -------------------- | ----- |
> |     NT-EBM     | 23.8M                | 78.12 |
> |      GLOW      | 44.2M                | 45.99 |
> |    EBM_FCE     | 44.9M                | 37.30 |
> |    CoopFlow    | 45.9M                | 21.16 |
> | CoopFlow(pre)  | 45.9M                | 15.80 |
> | Flow++ only \* | 28.8M                | 92.10 |
> | VAEBM          | 135.1M               | 12.16 |
>
>
>
> In Table 1.2, we also report the FID performance of different related baseline methods under the different numbers of MCMC steps. The two closely related baselines are (1) the single EBM with short-run MCMC starting from noise distribution [b], and (ii) Cooperative learning of an EBM and a generic generator [c]. We can see that with the same number of Langevin steps, the CoopFlow can generate much more realistic image patterns than the other two baselines. The quality of the generated images are evaluated by FID. The results of the experiments show that the proposed CoopFlow can use less number of MCMC steps to achieve better results. **We have put these results in the newly added subsection "A7. Analysis of the number of Langevin steps" in the Appendix.**
>
> Table 1.2: A comparison of FID scores under different numbers of MCMC steps.
>
> |            Method            | 10    | 20     | 30     | 40     | 50     | 200   |
> |:----------------------------:| ----- | ------ | ------ | ------ | ------ | ----- |
> | CoopFlow = EBM + flow (ours) | 16.46 | 15.20  | 15.80  | 16.80  | 15.64  | 17.94 |
> |   EBM with short-run MCMC    | 421.3 | 194.88 | 117.02 | 140.79 | 198.09 | 54.23 |
> |  CoopNet = EBM + generator   | 33.74 | 33.48  | 34.12  | 33.85  | 42.99  | 38.88 |
>
> =========== reference =======
>
> [a] NVAE:  A  deep  hierarchical  variational  autoencoder. NeurIPS, 2020.
>
> [b] Learning Non-Convergent Non-Persistent Short-Run MCMC Toward Energy-Based Model. NeurIPS 2019
>
> [c] Cooperative Training of Descriptor and Generator Networks. PAMI 2018.

---

> ### Author Response · Authors · 2021-11-23
> **Response to Reviewer QoKP (Part 3 of 8)**
>
> **Q4. About the analysis of the bias of the Contrastive divergence objective in Equation (8)**
>
> The EBM trained by contrastive divergence (CD) [a] still follows the "analysis by synthesis" scheme to update the model parameters. The learning gradient is still computed as Eq.(2) in our paper, i.e.,
>
> \begin{equation}
> L'(\theta)=E_{p_{\text{data}}}[\nabla_{\theta}f_{\theta}(x)]-E_{p_{\theta}}[\nabla_{\theta}f_{\theta}(x)]\approx \frac{1}{n}\sum_{i=1}^{n}\nabla_{\theta}f_{\theta}(x_i)-\frac{1}{n}\sum_{i=1}^{n}\nabla_{\theta}f_{\theta}(\tilde{x}_i),
> \end{equation}
>
> except that the synthesized examples $\tilde{x}$ is sampled by a short-run MCMC initialized by the training data. For example, the CD-1 is commonly used to train Restricted Boltzmann machines (a special EBM), and "1" means the single-step MCMC. We denote $\tilde{x} \sim K_{\theta} p_{data}(x)$, where $K_{\theta}$ is the MCMC transition kernel of $T$ steps of MCMC that samples $p_{\theta}$, and  $(K_{\theta} p_{data})(x)$ is the distribution of $x$ after $T$ steps of MCMC update starting from $q_{data}$. Because $\tilde{x} \sim K_{\theta} p_{data}(x) \neq p_{\theta}(x)$, therefore, the estimated $\theta$ using the above equation is biased (i.e., it is not an MLE solution, but it still a convergent algorithm with fixed point!). Hinton pointed out that the CD learning actually minimize the following objective, which is a perturbation of MLE,
>
> \begin{equation}
> \min_{\theta}
> D_{\text{KL}}(p_{\text{data}}||p_{\theta})-D_{\text{KL}}(K_{\theta} p_{data}|| p_{\theta} ),
> \end{equation}
> where the first term is the MLE, and the second term is the bias due to the gap between the actually used sampling distribution $K_{\theta} p_{data}$ and the correct distribution we should sample from, i.e., $p_{\theta}(x)$. Only if $D_{\text{KL}}(K_{\theta} p_{data}|| p_{\theta} )=0$, the CD estimate will be unbiased. However, the original CD doesn't explicitly minimize the second term. Recently, some work [a] improves the CD by explicitly minimizing the second term to further reduce the bias.
>
> We need to point out that, even though the CD is biased, it is still a convergent algorithm, which has been well studied in its original paper [a]. "Biased" doesn't mean it is wrong. "Biased" is just because $D_{\text{KL}}(K_{\theta} p_{data}|| p_{\theta} ) \neq0$. Actually, many useful algorithms and models using amortized sampling or amortized inference are biased (with respect to MLE). For example, VAE, whose objective is also a perturbation of MLE,
>
> $\min_{\alpha,\phi} D_{\text{KL}}(p_{\text{data}}(x)||p_{\alpha}(x)) + D_{\text{KL}}(q_{\phi}(z|x)||p_{\alpha}(z|x))$
>
> where $p_{\alpha}$ is the distribution of the decoder/generator in VAE, and $q_{\phi}(z|x)$ is the encoder to approximate the true posterior $p_{\alpha}(z|x)$. Even though VAE explicitly minimizes the second term (i.e., bias) in its loss, but in practice, this term can not be zero due to the mean-field design of the inference model $q_{\phi}$, the model $\alpha$ is still biased to MLE. Even though our model is not related to VAE, we just want to say "bias" is not a bad thing, it just a trade-off between computation efficiency and the exact MLE solution due to the usage of amortized sampling/inference.
>
> **(The reply of this question Q4 is unfinished and it is to be continued in the "Second part of reply for Q4")**

---

> > ### Author Response · Authors · 2021-11-23
> > **Second part of reply for Q4**
> >
> > **(This is the second part of reply for Q4)**
> >
> > Now let us go back to the CoopFlow. The update of the EBM in our CoopFlow can be considered **a modified CD** algorithm, because its MCMC starts from the normalizing flow $q_{\alpha}$. At iteration $t$, the update of the EBM follows the objective
> >
> > $\theta^{(t+1)} = \arg \min_{\theta}
> > D_{\text{KL}}(p_{\text{data}}||p_{\theta})-D_{\text{KL}}(K_{\theta^{(t)}} q_{\alpha^{(t)}}|| p_{\theta} )  ....... (i)$.
> >
> > This is biased  if $D_{\text{KL}}(K_{\theta^{(t)}} q_{\alpha^{(t)}}|| p_{\theta} )\neq0$.
> >
> > However, our algorithm has the other part that can handle the second term. It is the MLE learning of the normalizing flow $q_{\alpha}$, with $\tilde{x} \sim K_{\theta^{(t)}} q_{\alpha^{(t)}}$ as training examples. That is, at iteration $t$, we fix $K_{\theta^{(t)}} q_{\alpha^{(t)}}$ and minimize the following objective
> >
> > $\alpha^{(t+1)} = \arg \min_{\alpha} D_{\text{KL}}(K_{\theta^{(t)}} q_{\alpha^{(t)}}|| q_{\alpha} ).....(ii)$.
> >
> > Eq.(ii) tries to reduce the distance between the flow and the EBM. In an ideal case where capacities of all models are large enough and the MCMC $K_{\theta}$ is a convergent MCMC, minimizing Eq.(ii) will lead to  $D_{\text{KL}}(K_{\theta}q_{\alpha}|| q_{\alpha})\rightarrow 0$, which means $q_{\alpha}=p_{\theta}$ because $q_{\alpha}$ has become the stationary distribution of $K_{\theta}$. If $q_{\alpha}=p_{\theta}=0$, then the second term of Eq(i) will be vanishing because $D_{\text{KL}}(K_{\theta} q_{\alpha}|| p_{\theta} )=D_{\text{KL}}(K_{\theta} p_{\theta}|| p_{\theta} )=0$, the Eq.(i) only has the first term left, which is the MLE solution. Since $q_{\alpha}=p_{\theta}$, thus $\alpha$ is also MLE solution. Therefore, our estimates of $\theta$ and $\alpha$ in the CoopFlow are unbiased. **This has been analyzed in the paragraph "Ideal case analysis" in Section 4.2 in our original submission**.
> >
> > In practice, the conditions like large enough capacity of models and convergent MCMC are very hard to meet. Eventually, the CoopFlow might converge to a fixed point but not a fixed point of MLE. **We have used information geometry to illustrate the fixed point in Figure 1 and analyze the bias in the paragraph "moment matching estimator" of section 4.2.** That is, even though the CoopFlow is not MLE solution, but it is a moment matching estimator (MME), which is a perturbation of MLE shown in Figure 1. MLE is shown as $p_{\theta_{MLE}}$ in Figure 1, and the obtained MME solution is marked as a star $\pi^\star$. Again, bias doesn't means wrong. $\pi_{\theta,\alpha}$ with MME is a valid generator.
> >
> > **We have revised our paper and added a new subsection "A9. Discussion on the bias of the contrastive divergence objective for EBM" in the appendix to include a discussion about bias**. Please check the formal reply in the revised paper. Thank you so much for improving our paper.
> >
> > ===================reference=====================
> >
> > [a] G. Hinton. “Training Products of Experts by Minimizing Contrastive Divergence. 2022.
> >
> > [b] Improved Contrastive Divergence Training of Energy Based Models. ICML 2021.

---

> ### Author Response · Authors · 2021-11-23
> **Response to Reviewer QoKP (Part 4 of 8)**
>
> **Q5. The ability to model the density is not demonstrated empirically. They do not validate this property empirically even on a toy example.**
>
> We have demonstrated the ability of model the density (distribution) in both toy data and real image data in our original submission.
>
> As to the 2D toy data example in Section 5.1 and Figure 2, we have shown the results of the proposed CoopFlow in both the scenario using short-run non-convergent MCMC and the scenario using long-run convergent MCMC. All the results has verified the claims in our paper. For example, in the scenario using long-run convergent MCMC (green box with $T=10000$), both normalizing flow $q_{\alpha}$ and EBM $p_{\theta}$ will be maximum likelihood solution (This is explained in the paragraph "Ideal case analysis" in Section 4.2). Therefore the densities of both $p_{\theta}$ and $q_{\alpha}$ will be perfect match with the ground-truth data distribution. Additionally, in the scenario using short-run non-convergent MCMC, since the EBM is just a moment matching estimator, the EBM is not valid but the short-run Langevin flow serves as a flow-based correction of the normalizing flow, that is the Two-flow generator is valid. This has been verified by the result shown in the green box with $T=100$, in which the EBM $p_{\theta}$ is invalid but the CoopFlow $\pi_{\theta,\alpha}$ (including both normalizing flow and Langevin flow) is a valid generator that captures the ground-truth data distribution.
>
> In the real image data, since the data distribution is highly multi-modal, MCMC is hard and even impossible to converge. We focus on the short-run MCMC scenario and show that the CoopFlow is a valid generator by evaluating it via FID score, which is the golden standard of diagnoses of the distance between the learned density (distribution) and the target distribution.

---

> ### Author Response · Authors · 2021-11-23
> **Response to Reviewer QoKP (Part 5 of 8)**
>
> **Q6. No convergence analysis.**
>
> The CoopFlow algorithm simply involves two MLE learning algorithms: (i) the MLE learning of the EBM $p_{\theta}$ and (ii) the MLE learning of the normalizing flow $q_{\alpha}$. The convergence of each of the two learning algorithms has been well studied and verified in the existing literature, e.g., [a][b][c]. That is, each of them has a fixed point. The only interaction between these two algorithms in the proposed CoopFlow algorithm is that, in each iteration, they feed each other with their synthesized examples and use the cooperatively synthesized examples in their parameter update formulas. To be specific, the normalizing flow uses its synthesized examples to initialize the MCMC of the EBM, and the EBM feed the normalizing flow with its synthesized examples as training examples. The synthesized examples from the Langevin flow are considered the cooperatively synthesized examples by two models, and used to compute their learning gradients. Unlike other amortized sampling methods (e.g., [d][e]) that uses variational learning, the EBM and normalizing flow in our framework don't back-propagate each other through the cooperatively synthesized examples. They just feed each other with some input data for their own training algorithms. That is, each learning algorithm will still converge and have a fixed point.
>
> Now let us analyze the convergence of the whole CoopFlow algorithm that alternates two MLE algorithms. We first analyze the convergence of the objective function at each learning step, and then we conclude the convergence of the whole algorithm.
>
> **The convergence of CD learning of EBM**. The learning objective of the EBM is to minimize the KL distance between the EBM and the data distribution. Since the MCMC of EBM in our case is initialized by the normalizing flow, it follows a modified contrastive divergence algorithm. That is, at iteration $t$, it has the following objective,
> \begin{align}
> \theta^{(t+1)} = \arg \min_{\theta}
> D_{\text{KL}}(p_{\text{data}}||p_{\theta})-D_{\text{KL}}(K_{\theta^{(t)}} q_{\alpha^{(t)}}|| p_{\theta}).
> \end{align}
> No matter what kind of distribution is used to initialize the MCMC, it will has a fixed point when the learning gradient of $\theta$ equals to 0, i.e., $L'(\theta)= \frac{1}{n}\sum_{i=1}^{n}\nabla_{\theta}f_{\theta}(x_i)-\frac{1}{n}\sum_{i=1}^{n}\nabla_{\theta}f_{\theta}(\tilde{x}_i)=0$. The initialization of the MCMC only affects the location of the fixed point of the learning algorithm. The convergence and the analysis of the fixed point of contrastive divergence has been studied by papers [f][g].
>
> **The convergence of MLE learning of normalizing flow** The normalizing flow’s objective is to learn to minimize the KL divergence between the normalizing flow and the Langevin flow because, in each learning iteration, the normalizing flow uses the EBM’s synthesized examples as training data. At iteration $t$, it has the following objective
> \begin{align}
> \alpha^{(t+1)} = \arg \min_{\alpha} D_{\text{KL}}(K_{\theta^{(t)}} q_{\alpha^{(t)}}|| q_{\alpha} ),
> \end{align}
> which is a convergent algorithm at each $t$. The convergence has been studied by [a].
>
> **(The reply of this question Q6 is unfinished and it is to be continued in the "Second part of reply for Q6")**

---

> > ### Author Response · Authors · 2021-11-23
> > **second part of reply for Q6**
> >
> > **(this is the second part of reply for Q6**
> >
> > **The convergence of CoopFlow** The CoopFlow alternates the above two learning algorithms. The EBM learning will reduce the KL distance between EBM and data, i.e., $p_{\theta} \rightarrow p_{data}$; while the MLE learning of normalizing flow will reduce the KL distance between normalizing flow and EBM, i.e., $q_{\alpha} \rightarrow p_{\theta}$. Therefore, the normalizing flow will chase the EBM toward the data distribution gradually. Because the process $p_{\theta} \rightarrow p_{\text{data}}$ will stop at a fixed point, therefore $q_{\alpha} \rightarrow p_{\theta}$ will also stop at a fixed point obviously. Such a chasing game is a contraction algorithm, therefore the fixed point of CoopFlow exists. Empirical evidence has also supported that our EM-like learning algorithm is a contraction. If we use $(\theta^*,\alpha^*)$ to denote the fixed point of the CoopFlow, according to the definition of a fixed point, $(\theta^*,\alpha^*)$  will satisfy
> > \begin{align}
> > \theta^{\star}&= \arg \min_{\theta}D_{\text{KL}}(p_{\text{data}}||p_{\theta})-D_{\text{KL}}(K_{\theta^{\star}} q_{\alpha^{\star}}|| p_{\theta} ) \\\\
> > \alpha^{\star} &= \arg \min_{\alpha} D_{\text{KL}}(K_{\theta^{\star}} q_{\alpha^{\star}}|| q_{\alpha}).
> > \end{align}
> > The convergence of the cooperative learning framework (CoopNets) that integrates MLE algorithm of an EBM and MLE algorithm of a generic generator has been verified and discussed in the original paper of CoopNets [h]. The CoopFlow that uses a normalizing flow instead of a generic generator has the same convergence property as that of the original CoopNets. The major contribution in our paper is to start from the above fixed point equation to discuss where the fixed point will be in our algorithm when the MCMC of the EBM is non-mixing and non-convergent. All the existing cooperative learning papers do not study the scenario of short-run MCMC. Our paper goes beyond all the prior works about cooperative learning.
> >
> > **We have followed you suggestion to add the above convergence analysis in the newly added subsection "A8. More Convergence Analysis" in the appendix of our revision. Please take a look!**
> >
> > ================== reference ===================
> >
> > [a] Younes, L. 1999. On the convergence of markovian stochastic algorithms with rapidly decreasing ergodicity rates. Stochastics: An International Journal of Probability and Stochastic Processes 65(3-4):177–228.
> >
> > [b] Diederik P. Kingma. Glow: Generative Flow with Invertible 1x1 Convolutions.
> >
> > [c] A Theory of Generative ConvNet. ICML 2016
> >
> > [d] Divergence Triangle for Joint Training of Generator Model, Energy-based Model, and Inference Model. CVPR 2018
> >
> > [e] No MCMC for me: Amortized sampling for fast and stable training of energy-based models. ICLR 2021.
> >
> > [f] Geoffrey E. Hinton. Training products of experts by minimizing contrastive divergence. Neural Computation, 2002
> >
> > [g] Miguel and  Geoffrey  E.  Hinton. On  contrastive  divergence  learning.    AISTATS, 2005.
> >
> > [h] Xie et al. Cooperative Training of Descriptor and Generator Networks. TPAMI 2018.

---

> ### Author Response · Authors · 2021-11-23
> **Response to Reviewer QoKP (Part 6 of 8)**
>
> **Q7. About "an infinite looping" means "not converge". The non-convergence property is a significant flaw of the approach.**
>
> We have to say you have a **misunderstanding** of the term "infinite looping". The term "infinite looping" doesn't mean "non-convergence" in our statement. It just describes the behaviors of the CoopFlow learning algorithm in the manifold space when the algorithm arrives at a fixed point. Yes! I didn't say it wrong. It is an infinite looping at a fixed point! The term "fixed point" and "infinite looping" are not contradictory. Almost all the EM-type algorithm has this property.
>
> Let me first take the VAE as an example and illustrate its infinite looping at its fixed point. Let $x$ be data and $z$ be latent variables following prior distribution $p(z)$. We use $p_{\text{data}}(x)$ for data distribution, $p_{\alpha}(x|z)$ for the decoder, and $q_{\phi}(z|x)$ for the encoder.  Recall the objective of VAE
>
> \begin{align}
> &\min_{\alpha,\phi}D_{\text{KL}}(p_{\text{data}}(x)||p_{\alpha}(x)) + D_{\text{KL}}(q_{\phi}(z|x)||p_{\alpha}(z|x))\\\\
> =&\min_{\alpha,\phi} D_{\text{KL}}(p_{\text{data}}(x)q_{\phi}(z|x)||p_{\alpha}(x|z)p(z)))\\\\
> =&\min_{\alpha,\phi}D_{\text{KL}}(q_{\phi}(z,x)||p_{\alpha}(z,x))
> \end{align}
>
> Let $Q = \{q_{\phi}(z,x), \forall \phi\}$ and $P = \{p_{\alpha}(z,x), \forall \alpha\}$ be the two families of joint distributions. According to information geometry, each of $Q$ and $P$ can be visualized as a curve to represent the manifold. Each point on the curve is an estimate of the distribution. Minimizing $D_{\text{KL}}(q_{\phi}||p_{\alpha})$ over $\phi$ and $\alpha$ means that we find one point on curve $Q$ and one point on curve $P$ such that distance between two points is minimum. We want to say three facts:
> (1) the VAE objective has a minimum, which means the learning of VAE is a contraction.
> (2) In general, the minimum of $D_{\text{KL}}(q_{\phi}||p_{\alpha})$ might not be necessarily zero, that is, there is no intersection between curve $Q$ and curve $P$.
> (3) Minimizing $D_{\text{KL}}(q_{\phi}||p_{\alpha})$ can be accomplished by alternating projection, that is, starting from
> $p = p_0 \in P$, we project $p_0$ onto $Q$ by minimizing $D_{\text{KL}}(q||p)$ over $q \in Q$ to obtain $q_1$. Then we project $q = q_1$ onto $P$ by minimizing $D_{\text{KL}}(q||p)$ over $p \in P$ to obtain $p_1$, and so on. This process will converge to a local minimum of $D_{\text{KL}}(q||p)$. That is, let $(p^*,q^*)$ be the fixed point that obtains local minimum, they satisfy the fixed point property, i.e.,
> \begin{align}
> &q^*=\arg \min_{q}D_{\text{KL}}(q||p^*). \\\\
> &p^*=\arg \min_{p}D_{\text{KL}}(q^*||p).
> \end{align}
> Consider the fixed points $q^*$ in curve Q and $p^*$ in curve $P$. When you keep running the above projections (each minimization of KL is a projection process), $(q^*,p^*)$ will not changed any more. but the projection process between two curves still happens. Since there is no intersection between $Q$ and $P$. The mutual projections is like an infinite looping between points $q^*$ and $p^*$. For visualization, please check the Figure 2 (right hand side) of this paper https://openreview.net/pdf?id=NXbapYR49pg
>
> **(The reply of this question Q7 is unfinished and it is to be continued in the "Second part of reply for Q7")**

---

> > ### Author Response · Authors · 2021-11-23
> > **Second part of reply for Q7**
> >
> > **(This is the second part of reply for Q7")**
> >
> > Now, get back to our case. We believe your misunderstanding comes from this sentence in our paper: **"At convergence of the learning algorithm, there is an infinite looping, i.e., $K_{\theta^{\star}}q_{\alpha^{\star}}$ (i.e., $\pi^{\star}$) lifts $q_{\alpha^{\star}}$ off the ground $A$, and the projection drops $K_{\theta^{\star}}q_{\alpha^{\star}}$ back to $q_{\alpha^{\star}}$"** at the end of section 4.2 when we analyze the behavior of the learning algorithm using information geometry in Figure 1. We have shown in your question Q6 that the CoopFlow algorithm is a convergent algorithm, that is, there is a fixed point $(\theta^{\star},\alpha^{\star})$ that satisfies
> >
> > \begin{align}
> > \theta^{\star} &= \arg \min_{\theta}D_{\text{KL}}(p_{\text{data}}||p_{\theta})-D_{\text{KL}}(K_{\theta^{\star}} q_{\alpha^{\star}}|| p_{\theta})\\\\
> > \alpha^{\star} &= \arg \min_{\alpha} D_{\text{KL}}(K_{\theta^{\star}} q_{\alpha^{\star}}||q_{\alpha})\\\\
> > \end{align}
> >
> >
> > $q_{\alpha^{\star}}$ is a point in curve $A=\\{q_{\alpha}, \forall \alpha\\}$, and $\pi^{\star}=K_{\theta^{\star}}q_{\alpha^{\star}}$ is a point in the curve $\Omega=\\{p: E_p[h(x)]=E_{p_{data}}[h(x)]\\}$, where $h(x)=\nabla_{\theta}f_{\theta}(x)$. The distance between point $q_{\alpha^{\star}}$  and point  $\pi^{\star}$ is the minimum distance between curve $A$ and curve $\Omega$. When we keep running our algorithm at the fixed point, there is a looping between point $\pi^{\star}$ and point $q_{\alpha^{\star}}$.
> >
> > To be specific, the synthesis step starting from $q_{\alpha^{\star}}$ will get to $\pi^{\star}$ via short-run MCMC $K_{\theta^{\star}}$, and then the learning step $\alpha^{\star}=\min_{\alpha} D_{\text{KL}}(K_{\theta^{\star}} q_{\alpha^{\star}}||q_{\alpha})$ project $\pi^{\star}$ (i.e., $K_{\theta^{\star}}q_{\alpha^{\star}}$) back to the point $q_{\alpha^{\star}}$. There is a looping (MCMC shifting from $A$ to $\Omega$ + projection from $\Omega$ to $A$) between the fixed points $\pi^{\star}$ and $q_{\alpha^{\star}}$.
> >
> > Please read the paragraph "Moment matching estimator" in section 4.2 again and re-think about Figure 1. We hope our explanation have corrected your misunderstanding. Please let us know if you still have question. Please re-consider your rating. Thanks.

---

> ### Author Response · Authors · 2021-11-23
> **Response to Reviewer QoKP (Part 7 of 8)**
>
> **Q8. NT-EBM baseline on Celeba dataset**
>
> Thank you for pointing out this. This is a mistakenly reference here and we have deleted it in our modified version. For other scores, the VAE, DCGAN, GLOW, EBM-FCE scores are got from [a], which are a fair comparisons on 32x32 resolution CelebA. The score of GEBM[b] is also reported on the 32 x 32 downsampled version. On the other hand, our model outperforms NT-EBM (as well as a lot of other baseline models) by a large margin on the CIFAR-10 and SVHN datasets, which are also fair comparison on images with the same resolution.
>
> [a] Gao, Ruiqi, et al. "Flow contrastive estimation of energy-based models." Proceedings of the IEEE/CVF Conference on Computer Vision and Pattern Recognition. 2020.
>
> [b] Michael Arbel, Liang Zhou and Arthur Gretton.
> "Generalized energy based models." nternational Conference on Learning Representations, ICLR 2021.
>
> **Q9. Reconstruction use case is not clear. It is not clear what kind of a problem the authors approach there. Are these images corrupted by some noise?**
>
> The reconstruction case is used to verify the short-run MCMC is non-convergent, such that $K$-step Langevin flow actually a $K$-layer noise-injcetd residual network. Then the resulting CoopFlow (i.e., normalizing flow + Langevin flow) is a generator that maps the Gaussian noise $z$ to the image $x$, where the $z$ can be considered as latent variable.  So that it should be used for reconstruction. Please note that if the MCMC converges, then the MCMC  is impossibe to perform reconstruction, since the MCMC's output is independent to its starting point. Only short-run MCMC can be considered a determinstic transformation. The pioneering work that studies EBM with short-run MCMC also uses a reconstruction case to show that the non-convergent short-run MCMC is a valid flow-like generator by using it for reconstruction. In our paper, the CoopFlow in the short-run MCMC senario can be considered as a normalizng flow folowed by a short-run Langevin flow. This two-flow generator can be used for reconstruction purpose.
>
> The experiment shown in Figure 3 is a reconstruction of clean images, without corruption.
>
> **Q10. About the reconstrucion by CoopFlow, the choice of the intermediate representation (the output space of the flow) for the latent space is also not motivated. Why couldn't one take the initial distribution that is an input for the flow?**
>
> This is a very good question! Recall our model: $z \sim p_0(z); \hat{x}=g_{\alpha}(z); x=F_{\theta}(\hat{x},e)$, where $\theta$ is the Langevin flow, and $\alpha$ is normalizing flow, $p_0$ is Gaussian prior, and $e$ contains all the injected noises in the Langevin flow. (The reason why we can write the Langevin flow as a mapping $x=F_{\theta}(\hat{x},e)$ because it is a short-run non-convergent MCMC whose output $x$ is dependent on the initialization $\hat{x}$; otherwise $x$ will converge to the target distribution no matther where it starts. That is $x$ is independent on $\hat{x}$).
>
> The true latent space in the above model is indeed $z$. In general, we should infer $z$ for image reconstruction, that is, given an image $x$, we find its latent code $z$ by using gradient descent on $L(z)=||x-F_{\theta}(g_{\alpha}(z),e)||^2$, with $z$ is initialized by $p_0$. However, $g$ is an invertible transformation. therefore we can infer $z$ by an efficient way, i.e., we can first find $\hat{x}$ by gradient descent on $L(\hat{x})=||x-F_{\theta}(\hat{x},e)||^2$, and then use $z=g_{\alpha}^{-1}(\hat{x})$ to get $z$. These two methods are equivalent, but the latter on is computationally efficient, since computing the gradient on the whole two-flow generator $F_{\theta}(g_{\alpha}(z),e)$ is time consuming.
>
> We need to point out that if $g$ is not the normalizing flow but a generic generator, we have to use the frst method to infer $z$ for reconstruction. This is the advantage of using normalizing flow in our framework instead of other top-down generator. This might somehow answer your question about why choosing normaling flow, because its invertibility will make the inference step more efficient than other alternatives.

---

> ### Author Response · Authors · 2021-11-23
> **Response to Reviewer QoKP (Part 8 of 8)**
>
> **Q11. Numerous applications listed after the generative modeling experiments should be measured quantitatively rather than qualitatively by several examples.**
>
> Following your suggestion, we have add many new experiments and comparisons in our modified paper.
>
> (i) In the subsection "A4. More Generation Results" in the Appendix, we add the FID curve over training epochs to prove the convergence of our model;
>
> (ii) In the newly added subsection "A5.Quantitative Results For The Image Reconstruction Experiment" in the Appendix. We report the per pixel MSE for our image reconstruction experiment and show that CoopFLow does a better job than Short-run EBM.
>
> (iii) In the newly added subsection "A6. Comparison of model sizes" in the Appendix， we compare the model size and image generation quality (via FID scores) of different models and thus shows that our model is efficient in terms of model size.
>
> (iv) In the newly added subsection "A7.Analysis of the number of Langevin steps" in the Appendix, we compare models of different MCMC steps and show that our model can achieve better performance with less MCMC steps.

---

> ### Author Response · Authors · 2021-11-28
> **Looking forward to your feedback and discussion**
>
> Dear Reviewer QoKP,
>
> We thank you again for your valuable comments and suggestions to help us improve our paper in the aspects of presentation, experiments, analysis, and discussion.
>
> Our current revised paper has taken into account all your suggestions, please kindly let us know if you still have follow-up questions.
>
> Thanks,
>
> Authors

---

> ### Comment · Reviewer_QoKP · 2021-11-29
> **Rebuttal answer**
>
> Thank you for the clarifications.
> In my opinion, writing 8 parts (3 of which consist of 2 pieces) of rebuttal is beyond the reasonable limit for clarifications.
>
> 2] I agree that combining different approaches could potentially alleviate their downsides. However, this should be tested and demonstrated empirically. For example, in the experimental section that the proposed algorithm achieves comparable results to SOTA EBMs. Does it mean that we do not need flows to achieve SOTA in the generative modeling? To answer this question, the proposed model should be compared against the corresponding EBM and the flow with the same computational budget (number of parameters, as chosen by the authors).
>
> 3] The list provided for the comparison is selective. I'm convinced that the authors could provide the comparison against the full list of the models from the experimental section (I would suggest using the scatter plot for this purpose). In the paper's context, the most interesting comparison will be against the models with better FID, which do not go beyond a single approach for generative modeling.
>
> 4.1] I don't understand what the words ```"Biased" doesn't mean it is wrong.``` mean because I'm not aware of the mathematical definition of "wrong."
> 4.2] Currently, it is impossible to understand from the main body of the paper whether the authors use the biased approximation of their loss, or they propose backpropagating through the samples $\widetilde{x}_i$.
>
> 5] According to the captions, Figure 2 demonstrates the learned density of the flow part of the model and of the EBM, but not of the joint model (which is the proposed model).
>
> 6] The discussion proposed by the authors is not a formal proof of algorithm convergence. Using the same reasoning one can conclude that GANs training also converges.
>
> 7] I find the usage of such wording as "infinite looping at convergence" to be confusing. It might be confused with the convergence to the stable orbit of a dynamical system (as might happen during GANs training, for instance). However, I leave it up to the authors.
>
> 9] Thank you for the clarification.
>
> 10] Thank you for clarifying this point in the text.
>
> I have gone through the rebuttal. Unfortunately, my main concerns are not addressed. Therefore, I would like to keep my initial assessment of the paper.

---

> > ### Author Response · Authors · 2021-11-29
> > **Reply to "Reviewer QoKP's Reply to the rebuttal" (Part 1 of 2)**
> >
> > **Q0: In my opinion, writing 8 parts (3 of which consist of 2 pieces) of rebuttal is beyond the reasonable limit for clarifications.**
> >
> > According to the ICLR's rule, there is no limit for ICLR's rebuttal. This is because ICLR's openreview scheme encourages us to have a thorough communication with the reviewers. We do respect and take serious of each word from each reviewer.
> >
> >
> > **Q1: I agree that combining different approaches could potentially alleviate their downsides. However, this should be tested and demonstrated empirically. For example, in the experimental section that the proposed algorithm achieves comparable results to SOTA EBMs. Does it mean that we do not need flows to achieve SOTA in the generative modeling? To answer this question, the proposed model should be compared against the corresponding EBM and the flow with the same computational budget (number of parameters, as chosen by the authors).**
> >
> >
> > 1. As to flow with the same number of parameters: In the Table 8 of our paper, we can see that comparing with other flow based models (e.g., GLOW and EBM-FCE) with the similar number of parameters, our model achieve the best FID. Note that GLOW is a pure normalizing flow model, while EBM-FCE is an EBM + flow framework.
> >
> > Table 8
> >
> > |     Method     | number of parameters | FID   |
> > |:--------------:| -------------------- | ----- |
> > |     NT-EBM     | 23.8M                | 78.12 |
> > |      GLOW      | 44.2M                | 45.99 |
> > |    EBM_FCE     | 44.9M                | 37.30 |
> > |    CoopFlow    | 45.9M                | 21.16 |
> > | CoopFlow(pre)  | 45.9M                | 15.80 |
> > | Flow++ only \* | 28.8M                | 92.10 |
> > | VAEBM          | 135.1M               | 12.16 |
> >
> > 2. As to EBM, we want to point out that the motivation of our paper is to amortize the sampling to the flow model to reduce the number of the Langevin steps. Adding a flow model for EBM indeed increases the parameters. However, it also greatly reduces the number of MCMC steps needs during sampling. The number of MCMC steps is also computational budge (not just parameters). Table 10 below (which has been in our paper) has shown that CoopFlow is more efficient compared with other models with the same computational budges (number of MCMC steps). The conclusion is that, with the same computational budge, our method is best shown in Table 10.
> >
> >
> >
> > Table 10: A comparison of FID scores under different numbers of MCMC steps.
> >
> > |            Method            | 10    | 20     | 30     | 40     | 50     | 200   |
> > |:----------------------------:| ----- | ------ | ------ | ------ | ------ | ----- |
> > | CoopFlow = EBM + flow (ours) | 16.46 | 15.20  | 15.80  | 16.80  | 15.64  | 17.94 |
> > |   EBM with short-run MCMC    | 421.3 | 194.88 | 117.02 | 140.79 | 198.09 | 54.23 |
> > |  CoopNet = EBM + generator   | 33.74 | 33.48  | 34.12  | 33.85  | 42.99  | 38.88 |
> >
> > We think these evidences have demonstrated the advantages of our models. We don't claim that "we do not need flows to achieve SOTA". We just want to say the EBM can use its MCMC to improve the expressivity of the flow model, while the flow can help reduce the number of MCMC steps of EBM. The combination can lead to the best of both worlds.
> >
> >
> > **Q2: The list provided for the comparison is selective. I'm convinced that the authors could provide the comparison against the full list of the models from the experimental section (I would suggest using the scatter plot for this purpose). In the paper's context, the most interesting comparison will be against the models with better FID, which do not go beyond a single approach for generative modeling.**
> >
> >
> > We will follow your suggestion to use the scatter plot for demonstrating the comparison. The list only select and compare related baselines, which mainly use EBM and flow. There is no need to compare with other types of generative methods in terms of number of parameters. Again, we are not trying to propose a STOA generative model. We are studying to train an EBM with both short-run MCMC and amortized sampling. The short-run MCMC is an underexplored topic. **It is not reasonable to reject a paper just because it is not STOA.**
> >
> >
> >
> >
> > **Q3: I don't understand what the words "Biased" doesn't mean it is wrong. mean because I'm not aware of the mathematical definition of "wrong."**
> >
> > We juts wanted to point out that the Contrastive divergence and variational learning are biased, but they are not wrong models. "Biased" is just relative to the maximum likelihood estimator (MLE). There are still other non-MLE estimators, for example, the moment matching estimator (MME), which is also biasd to MLE. Our CoopFlow with short-run and non-convergent MCMC is MME.

---

> > ### Author Response · Authors · 2021-11-29
> > **Reply to "Reviewer QoKP's Reply to the rebuttal" (Part 2 of 2)**
> >
> > **Q4: Currently, it is impossible to understand from the main body of the paper whether the authors use the biased approximation of their loss, or they propose backpropagating through the samples.**
> >
> > The algorithm 1 of our paper has clearly shown that we do **NOT** perform backpropagating through the samples. The cooperative learning alternates the MLE learning of the EBM and the MLE learning of the flow. The non-mixing MCMC will lead to a biased but valid moment matching estimation of the model.
> >
> > **If you really read our rebuttal carefully and you will find the answer in the following statement that has been highlighted in our paper:**
> >
> > "Unlike other amortized sampling methods (e.g., Han et al. (2019); Grathwohl et al. (2021)) that uses variational learning, the EBM and normalizing flow in our framework **don’t back-propagate** each other through the cooperatively synthesized examples. They just feed each other with some input data for their own training algorithms."
> >
> >
> > **Q5: According to the captions, Figure 2 demonstrates the learned density of the flow part of the model and of the EBM, but not of the joint model (which is the proposed model).**
> >
> > The joint model, i.e., CoopFlow, is shown in **CoopFlow samples ($\pi_{\theta,\alpha}$)** in Figure 2, which have verified the success of our model. We can change the visualization by following your suggestion. It is just a matter of visualization. **It is unreasonable to reject a paper just because we use a visualization method that you don't want**.
> >
> > **Q6: The discussion proposed by the authors is not a formal proof of algorithm convergence. Using the same reasoning one can conclude that GANs training also converges.**
> >
> > The proposed CoopFlow is built on an exisiting cooperative learning framework [a,b,c] that are used for joint training an EBM and a generator. The normalizng flow in our framework is just a special generator. Analyzing the convergence of an exisiting learning scheme is out of scope of this paper. And we do show that in our revision, the cooperative learning is based on the MLE algorithm of EBM and MLE algorithm of flow, both of them are well-known convergent algorithm. There is no need to prove the convergence of both MLE algorithms. Our paper's contribution is to show that the cooperative learning with short-run non-mixing MCMC will lead to a stochastic normalizing flow (Langevin flow+ normalizing flow) whose estimation is actually a moment matching estimator.
> >
> > We start from the convergence equations (Eq.8 and 9 in our paper) which are from [a] and study, under the scenario of non-mixing MCMC, where the convergent point will be. **It is unreasonable to reject the paper because it doesn't prove the convergence of a prior work that has been published in a premier journal paper or even a text book.**
> >
> > [a] Xie et al. Cooperative Training of Descriptor and Generator Networks. (IEEE Transactions on Pattern Analysis and Machine Intelligence) 2018. https://arxiv.org/pdf/1609.09408.pdf
> >
> > [b] Adrian Barbu, Song-Chun Zhu. Monte Carlo Methods. Springer 2020. https://bit.ly/mc_book_sample
> >
> > [c] Xie et al. Cooperative Training of Fast Thinking Initializer and Slow Thinking Solver for Multi-Modal Conditional Learning. (IEEE Transactions on Pattern Analysis and Machine Intelligence) https://arxiv.org/pdf/1902.02812.pdf
> >
> >
> > **Q7: I find the usage of such wording as "infinite looping at convergence" to be confusing. It might be confused with the convergence to the stable orbit of a dynamical system (as might happen during GANs training, for instance). However, I leave it up to the authors.**
> >
> > We can follow your suggestion to change the wording to avoid the confusion.
> >
> > -------------------
> > We don't think the above concerns are strong reason to reject a paper. They are just visualization issue, plot issue, STOA issue, or even an issue regarding the convergence of a prior work.  The major contribution of this work is to study the short-run non-mixing MCMC in the context of EBM cooperative learning. Actually, the information geometry interpretation of the model is more important than the numerical performance of this work. We hope you can re-consider your rating again.

---

### Author Response · Authors · 2021-11-24
**A summary of the major revision made in the current paper during rebuttal**

To all reviewers, ACs, and PCs:

We have tried our best to address all the concerns raised by all 4 reviewers by providing **detailed responses** to each of them. Besides, we have also **substantially revised** our paper by following all valuable suggestions from the reviewers.

The following is a summary to list the major revision we have made in our paper. And all the revised and newly added contents have been highlighted in red color in our uploaded paper for your convenient inspection.

1. We highlighted the motivation in the last paragraph of the Section 1 in our paper.

2. We added missing related references suggested by reviewers.

3.  We improved the presentation of "Section 5.3 Image Reconstruction" to highlight the purpose of the experiment, and added a discussion of an alternative method for image reconstruction.

4. We added quantitative results for the experiment of image reconstruction in  section "A5 Quantitative Results for the Image Reconstruction Experiment" in Appendix.

5. We added Figure 12 in Appendix to show evidence of the convergence of our method.

6. We added a new subsection "A.8 More Convergence Analysis" in Appendix to discuss the convergence of the algorithm.

7. We added a new subsection "A.9 Discussion on the bias of the Contrastive Divergence Objective for EBM" in Appendix to discuss the bias of the contrastive divergence and how this affect our model.

8. We added a new subsection "A.10 Comparison between CoopFlow and Short-run EBM via Information Geometry" in Appendix to illustrate a comparison between our method and the EBM with a short-run MCMC only. We demonstrate that our method with the normalizing flow can result in shorter-run MCMC.

9. We added a new subsection "A.6 Comparison of Model Sizes" in Appendix to compare both FID (i.e., synthesis performance) and model size (i.e., number of parameters).

10. We added a new subsection "A.7 Analysis of the number of Langevin Steps" in Appendix to compare performance of different methods under the same number of Langevin steps. We find that our method can get competitive results with less Langevin steps.

**We hope all reviewers can carefully read our detailed feedbacks that have addressed your concerns and re-consider your score to give this paper a chance**.

Thanks again.

Authors

---

### Decision · Program_Chairs · 2022-01-20

**Decision:**

Accept (Poster)

**Comment:**

The paper proposes a methodological improvement in the Langevin-based training of energy-based models. The idea is to initialize the Langevin flow used to train an energy-based model with a normalizing flow which learns to mimic the Langevin flow as the energy-based model is being trained. The method is empirically evaluated on synthetic data and image benchmarks.

The reviewers are currently divided: one argues for strong rejection, two for weak acceptance, and one for strong acceptance. In summary, the reviewers have expressed two main points of criticism: (a) that the motivation is unclear or not experimentally demonstrated; (b) that the convergence properties of the algorithm are unclear. Regarding (a), I believe the authors have adequately addressed this concern, and in my judgement the method is sufficiently motivated. Regarding (b), the authors responses have mostly relied on non-rigorous argumentation and appeal to prior work, so I don't think the issue has been addressed to the reviewers' satisfaction. Having said that, in my judgement lack of clarity regarding convergence is not a sufficient reason to reject the paper, as there don't seem to be reasonable doubts that the method doesn't converge in practice.

On balance, although the paper has certain weaknesses, it proposes an interesting and potentially useful method without major technical inadequacies, so I'm leaning towards recommending acceptance.